# Multi-Swap $k$-Means++

**Lorenzo Beretta**[*]
University of Copenhagen
lorenzo2beretta@gmail.com

**Vincent Cohen-Addad**
Google Research
cohenaddad@google.com

**Silvio Lattanzi**
Google Research
silviol@google.com

**Nikos Parotsidis**
Google Research
nikosp@google.com

## Abstract

The $k$-means++ algorithm of Arthur and Vassilvitskii (SODA 2007) is often the practitioners' choice algorithm for optimizing the popular $k$-means clustering objective and is known to give an $O(\log k)$-approximation in expectation. To obtain higher quality solutions, Lattanzi and Sohler (ICML 2019) proposed augmenting $k$-means++ with $O(k \log \log k)$ local search steps obtained through the $k$-means++ sampling distribution to yield a $c$-approximation to the $k$-means clustering problem, where $c$ is a large absolute constant. Here we generalize and extend their local search algorithm by considering larger and more sophisticated local search neighborhoods hence allowing to swap multiple centers at the same time. Our algorithm achieves a $9 + \varepsilon$ approximation ratio, which is the best possible for local search. Importantly we show that our approach yields substantial practical improvements, we show significant quality improvements over the approach of Lattanzi and Sohler (ICML 2019) on several datasets.

## 1  Introduction

Clustering is a central problem in unsupervised learning. In clustering one is interested in grouping together "similar" object and separate "dissimilar" one. Thanks to its popularity many notions of clustering have been proposed overtime. In this paper, we focus on metric clustering and on one of the most studied problem in the area: the Euclidean $k$-means problem.

In the Euclidean $k$-means problem one is given in input a set of points $P$ in $\mathbb{R}^d$. The goal of the problem is to find a set of $k$ centers so that the sum of the square distances to the centers is minimized. More formally, we are interested in finding a set $C$ of $k$ points in $\mathbb{R}^d$ such that $\sum_{p \in P} \min_{c \in C} ||p - c||^2$, where with $||p - c||$ we denote the Euclidean distance between $p$ and $c$.

The $k$-means problem has a long history, in statistics and operations research. For Euclidean $k$-means with running time polynomial in both $n, k$ and $d$, a 5.912-approximation was recently shown in Cohen-Addad et al. [2022a], improving upon Kanungo et al. [2004], Ahmadian et al. [2019], Grandoni et al. [2022] by leveraging the properties of the Euclidean metric. In terms of lower bounds, the first to show that the high-dimensional $k$-means problems were APX-hard were Guruswami and Indyk [2003], and later Awasthi et al. [2015] showed that the APX-hardness holds even if the centers can be placed arbitrarily in $\mathbb{R}^d$. The inapproximability bound was later slightly improved by Lee et al. [2017] until the recent best known bounds of Cohen-Addad and Karthik C. S. [2019], Cohen-Addad et al. [2022d, 2021] that showed that it is NP-hard to achieve a better than 1.06-approximation and hard to approximate it better than 1.36 assuming a stronger conjecture. From a more practical point of view, Arthur and Vassilvitskii [2009] showed that the widely-used popular heuristic of Lloyd Lloyd

---

[*]Authors are ordered in alphabetical order.

[1957] can lead to solutions with arbitrarily bad approximation guarantees, but can be improved by a simple seeding strategy, called $k$-means++, so as to guarantee that the output is within an $O(\log k)$ factor of the optimum Arthur and Vassilvitskii [2007].

Thanks to its simplicity $k$-means++ is widely adopted in practice. In an effort to improve its performances Lattanzi and Sohler [2019], Choo et al. [2020] combine $k$-means++ and local search to efficiently obtain a constant approximation algorithm with good practical performance. These two studies show that one can use the $k$-means++ distribution in combination with a local search algorithm to get the best of both worlds: a practical algorithm with constant approximation guarantees.

However, the constant obtained in Lattanzi and Sohler [2019], Choo et al. [2020] is very large (several thousands in theory) and the question as whether one could obtain a practical algorithm that would efficiently match the $9 + \varepsilon$-approximation obtained by the $n^{O(d/\epsilon)}$ algorithm of Kanungo et al. [2004] has remained open. Bridging the gap between the theoretical approach of Kanungo et al. [2004] and $k$-means++ has thus been a long standing goal.

**Our Contributions.** We make significant progress on the above line of work.

- We adapt techniques from the analysis of Kanungo et al. [2004] to obtain a tighter analysis of the algorithm in Lattanzi and Sohler [2019]. In particular in Corollary 4, we show that their algorithm achieves an approximation of ratio of $\approx 26.64$.

- We extend this approach to multi-swaps, where we allow swapping more than one center at each iteration of local search, improving significantly the approximation to $\approx 10.48$ in time $O(nd \cdot poly(k))$.

- Leveraging ideas from Cohen-Addad et al. [2021], we design a better local search swap that improves the approximation further to $9 + \varepsilon$ (see Theorem 12). This new algorithm matches the $9 + \varepsilon$-approximation achieved by the local search algorithm in Kanungo et al. [2004], but it is significantly more efficient. Notice that 9 is the best approximation achievable through local search algorithms, as proved in Kanungo et al. [2004].

- We provide experiments where we compare against $k$-means++ and Lattanzi and Sohler [2019]. We study a variant of our algorithm that performs very competitively with our theoretically sound algorithm. The variant is very efficient and still outperforms previous work in terms of solution quality, even after the standard postprocessing using Lloyd.

**Additional Related Work.** We start by reviewing the approach of Kanungo et al. [2004] and a possible adaptation to our setting. The bound of $9 + \varepsilon$ on the approximation guarantee shown by Kanungo et al. [2004] is for the following algorithm: Given a set $S$ of $k$ centers, if there is a set $S^+$ of at most $2/\varepsilon$ points in $\mathbb{R}^d$ together with a set $S^-$ of $|S^+|$ points in $S$ such that $S \setminus S^- \cup S^+$ achieves a better $k$-means cost than $S$, then set $S := S \setminus S^- \cup S^+$ and repeat until convergence. The main drawback of the algorithm is that it asks whether there exists a set $S^+$ of points in $\mathbb{R}^d$ that could be swapped with elements of $S$ to improve the cost. Identifying such a set, even of constant size, is already non-trivial. The best way of doing so is through the following path: First compute a coreset using the state-of-the-art coreset construction of Cohen-Addad et al. [2022b] and apply the dimensionality reduction of Becchetti et al. [2019], Makarychev et al. [2019], hence obtaining a set of $\tilde{O}(k/\varepsilon^4)$ points in dimension $O(\log k/\varepsilon^2)$. Then, compute grids using the discretization framework of Matousek [2000] to identify a set of $\varepsilon^{-O(d)} \sim k^{O(\varepsilon^{-2} \log(1/\varepsilon))}$ grid points that contains nearly-optimum centers. Now, run the local search algorithm where the sets $S^+$ are chosen from the grid points by brute-force enumeration over all possible subsets of grid points of size at most, say $s$. The running time of the whole algorithm with swaps of magnitude $s$, i.e.: $|S^+| \leq s$, hence becomes $k^{O(s \cdot \varepsilon^{-2} \log(1/\varepsilon))}$ for an approximation of $(1 + \varepsilon)(9 + 2/s)$, meaning a dependency in $k$ of $k^{O(\varepsilon^{-3} \log(1/\varepsilon))}$ to achieve a $9 + \varepsilon$-approximation. Our results improves upon this approach in two ways: (1) it improves over the above theoretical bound and (2) does so through an efficient and implementable, i.e.: practical, algorithm.

Recently, Grunau et al. [2023] looked at how much applying a greedy rule on top of the $k$-means++ heuristic improves its performance. The heuristic is that at each step, the algorithm samples $\ell$ centers and only keeps the one that gives the best improvement in cost. Interestingly the authors prove that from a theoretical standpoint this heuristic does not improve the quality of the output. Local search algorithms for $k$-median and $k$-means have also been studied by Gupta and Tangwongsan [2008] who drastically simplified the analysis of Arya et al. [2004]. Cohen-Addad and Schwiegelshohn [2017]

demonstrated the power of local search for stable instances. Friggstad et al. [2019], Cohen-Addad et al. [2019] showed that local search yields a PTAS for Euclidean inputs of bounded dimension (and doubling metrics) and minor-free metrics. Cohen-Addad [2018] showed how to speed up the local search algorithm using $kd$-trees (i.e.: for low dimensional inputs).

For fixed $k$, there are several known approximation schemes, typically using small coresets Becchetti et al. [2019], Feldman and Langberg [2011], Kumar et al. [2010]. The state-of-the-art approaches are due to Bhattacharya et al. [2020], Jaiswal et al. [2014]. The best known coreset construction remains Cohen-Addad et al. [2022c,b].

If the constraint on the number of output centers is relaxed, then we talk about bicriteria approximations and $k$-means has been largely studied Bandyapadhyay and Varadarajan [2016], Charikar and Guha [2005], Cohen-Addad and Mathieu [2015], Korupolu et al. [2000], Makarychev et al. [2016].

## 2 Preliminaries

**Notation.** We denote with $P \subseteq \mathbb{R}^d$ the set of input points and let $n = |P|$. Given a point set $Q \subseteq P$ we use $\mu(Q)$ to denote the mean of points in $Q$. Given a point $p \in P$ and a set of centers $A$ we denote with $A[p]$ the closest center in $A$ to $p$ (ties are broken arbitrarily). We denote with $\mathcal{C}$ the set of centers currently found by our algorithm and with $\mathcal{O}^*$ an optimal set of centers. Therefore, given $p \in P$, we denote with $\mathcal{C}[p]$ and $\mathcal{O}^*[p]$ its closest ALG-center and OPT-center respectively. We denote by $\texttt{cost}(Q, A)$ the cost of points in $Q \subseteq P$ w.r.t. the centers in $A$, namely

$$\texttt{cost}(Q, A) = \sum_{q \in Q} \min_{c \in A} ||q - c||^2.$$

We use ALG and OPT as a shorthand for $\texttt{cost}(P, \mathcal{C})$ and $\texttt{cost}(P, \mathcal{O}^*)$ respectively. When we sample points proportionally to their current cost (namely, sample $q$ with probability $\texttt{cost}(q, \mathcal{C}) / \texttt{cost}(P, \mathcal{C})$) we call this the $D^2$ distribution. When using $O_\varepsilon(\cdot)$ and $\Omega_\varepsilon(\cdot)$ we mean that $\varepsilon$ is considered constant. We use $\widetilde{O}(f)$ to hide polylogarithmic factors in $f$. The following lemma is folklore.

**Lemma 1.** *Given a point set $Q \subseteq P$ and a point $p \in P$ we have*

$$cost(Q, p) = cost(Q, \mu(Q)) + |Q| \cdot ||p - \mu(Q)||^2.$$

Let $O_i^*$ be an optimal cluster, we define the *radius* of $O_i^*$ as $\rho_i$ such that $\rho_i^2 \cdot |O_i^*| = \texttt{cost}(O_i^*, o_i)$, where $o_i = \mu(O_i^*)$. We define the $\delta$-*core* of the optimal cluster $O_i^*$ as the set of points $p \in O_i^*$ that lie in a ball of radius $(1 + \delta)\rho_i$ centered in $o_i$. In symbols, $\texttt{core}(O_i^*) = P \cap B(o_i, (1 + \delta)\rho_i)$. Throughout the paper, $\delta$ is always a small constant fixed upfront, hence we omit it.

**Lemma 2.** *Let $O_i^*$ be an optimal cluster and sample $q \in O_i^*$ according to the $D^2$-distribution restricted to $O_i^*$. If $cost(O_i^*, \mathcal{C}) > (2 + 3\delta) \cdot cost(O_i^*, o_i)$ then $\Pr[q \in core(O_i^*)] = \Omega_\delta(1)$.*

*Proof.* Define $\alpha := \texttt{cost}(O_i^*, \mathcal{C}) / \texttt{cost}(O_i^*, o_i) > 2 + 3\delta$. Thanks to Lemma 1, for each $c \in \mathcal{C}$ we have $||c - o_i||^2 \geq (\alpha - 1)\rho_i^2$. Therefore, for each $y \in \texttt{core}(O_i^*)$ and every $c \in \mathcal{C}$ we have

$$\texttt{cost}(y, c) = ||y - c||^2 \geq \left(\sqrt{\alpha - 1} - (1 + \delta)\right)^2 \cdot \rho_i^2 = \Omega_\delta(\alpha \rho_i^2).$$

Moreover, by a Markov's inequality argument we have $|O_i^* \setminus \texttt{core}(O_i^*)| \leq \frac{1}{1+\delta} \cdot |O_i^*|$ and thus $|\texttt{core}(O_i^*)| \geq \Omega_\delta(|O_i^*|)$. Combining everything we get

$$\texttt{cost}(\texttt{core}(O_i^*), \mathcal{C}) \geq |\texttt{core}(O_i^*)| \cdot \min_{\substack{c \in \mathcal{C} \\ y \in \texttt{core}(O_i^*)}} \texttt{cost}(y, c) = \Omega_\delta(|O_i^*|) \cdot \Omega_\delta(\alpha \rho_i^2)$$

and $|O_i^*| \cdot \alpha \rho_i^2 = \texttt{cost}(O_i^*, \mathcal{C})$, hence $\texttt{cost}(\texttt{core}(O_i^*), \mathcal{C}) = \Omega_\delta(\texttt{cost}(O_i^*, \mathcal{C}))$. □

## 3 Multi-Swap $k$-Means++

The single-swap local search (SSLS) $k$-means++ algorithm in Lattanzi and Sohler [2019] works as follows. First, $k$ centers are sampled using $k$-means++ (namely, they are sampled one by one according to the $D^2$ distribution, updated for every new center). Then, $O(k \log \log k)$ steps of local search follow. In each local search step a point $q \in P$ is $D^2$-sampled, then let $c$ be the center among

the current centers $\mathcal{C}$ such that $\texttt{cost}(P, (\mathcal{C} \setminus \{c\}) \cup \{q\})$ is minimum. If $\texttt{cost}(P, (\mathcal{C} \setminus \{c\}) \cup \{q\}) < \texttt{cost}(P, \mathcal{C})$ then we swap $c$ and $q$, or more formally we set $\mathcal{C} \leftarrow (\mathcal{C} \setminus \{c\}) \cup \{q\}$.

We extend the SSLS so that we allow to swap multiple centers simultaneously and call this algorithm multi-swap local search (MSLS) $k$-means++. Swapping multiple centers at the same time achieves a lower approximation ratio, in exchange for a higher time complexity. In this section, we present and analyse the $p$-swap local search (LS) algorithm for a generic number of $p$ centers swapped at each step. For any constant $\delta > 0$, we obtain an approximation ratio ALG/OPT $= \eta^2 + \delta$ where

$$\eta^2 - (2 + 2/p)\eta - (4 + 2/p) = 0. \tag{1}$$

**The Algorithm.** First, we initialize our set of centers using $k$-means++. Then, we run $O(ndk^{p-1})$ local search steps, where a local search step works as follows. We $D^2$-sample a set $In = \{q_1 \dots q_p\}$ of points from $P$ (without updating costs). Then, we iterate over all possible sets $Out = \{c_1 \dots c_p\}$ of $p$ distinct elements in $\mathcal{C} \cup In$ and select the set $Out$ such that performing the swap $(In, Out)$ maximally improves the cost[2]. If this choice of $Out$ improves the cost, then we perform the swap $(In, Out)$, else we do not perform any swap for this step.

**Theorem 3.** *For any $\delta > 0$, the $p$-swap local search algorithm above runs in $\widetilde{O}(ndk^{2p})$ time and, with constant probability, finds an $(\eta^2 + \delta)$-approximation of $k$-means, where $\eta$ satisfies Equation* (1).

Notice that the SSLS algorithm of Lattanzi and Sohler [2019] is exactly the $p$-swap LS algorithm above for $p = 1$.

**Corollary 4.** *The single-swap local search in Lattanzi and Sohler [2019], Choo et al. [2020] achieves an approximation ratio $< 26.64$.*

**Corollary 5.** *For $p = O(1)$ large enough, multi-swap local search achieves an approximation ratio $< 10.48$ in time $O(nd \cdot poly(k))$.*

### 3.1 Analysis of Multi-Swap $k$-means++

In this section we prove Theorem 3. Our main stepping stone is the following lemma.

**Lemma 6.** *Let ALG denote the cost at some point in the execution of MSLS. As long as ALG/OPT $> \eta^2 + \delta$, a local search step improves the cost by a factor $1 - \Omega(1/k)$ with probability $\Omega(1/k^{p-1})$.*

*Proof of Theorem 3.* First, we show that $O(k^p \log \log k)$ local steps suffice to obtain the desired approximation ratio, with constant probability. Notice that a local search step can only improve the cost function, so it is sufficient to show that the approximation ratio is achieved at some point in time. We initialize our centers using $k$-means++, which gives a $O(\log k)$-approximation in expectation. Thus, using Markov's inequality the approximation guarantee $O(\log k)$ holds with arbitrary high constant probability. We say that a local-search step is *successful* if it improves the cost by a factor of at least $1 - \Omega(1/k)$. Thanks to Lemma 6, we know that unless the algorithm has already achieved the desired approximation ratio then a local-search step is successful with probability $\Omega(1/k^{p-1})$. To go from $O(\log k)$ to $\eta^2 + \delta$ we need $O(k \log \log k)$ successful local search steps. Standard concentration bounds on the value of a Negative Binomial random variable show that, with high probability, the number of trial to obtain $O(k \log \log k)$ successful local-search steps is $O(k^p \log \log k)$. Therefore, after $O(k^p \log \log k)$ local-search steps we obtain an approximation ratio of $\eta^2 + \delta$.

To prove the running time bound it is sufficient to show that a local search step can be performed in time $\widetilde{O}(ndk^{p-1})$. This is possible if we maintain, for each point $x \in P$, a dynamic sorted dictionary[3] storing the pairs $(\texttt{cost}(x, c_i), c_i)$ for each $c_i \in \mathcal{C}$. Then we can combine the exhaustive search over all possible size-$p$ subsets of $\mathcal{C} \cup In$ and the computation of the new cost function using time $O(ndk^{p-1} \log k)$. To do so, we iterate over all possible size-$(p - 1)$ subsets $Z$ of $\mathcal{C} \cup In$ and update all costs as if these centers were removed, then for each point $x \in P$ we compute how much its cost increases if we remove its closest center $c_x$ in $(\mathcal{C} \cup In) \setminus Z$ and charge that amount to $c_x$. In the end, we consider $Out = Z \cup \{c\}$ where $c$ is the cheapest-to-remove center found in this way. $\qquad\square$

The rest of this section is devoted to proving Lemma 6. For convenience, we prove that Lemma 6 holds whenever ALG/OPT $> \eta^2 + O(\delta)$, which is wlog by rescaling $\delta$. Recall that we now focus on

---

[2]If $In \cap Out \neq \emptyset$ then we are actually performing the swap $(In \setminus Out, Out \setminus In)$ of size $< p$.

[3]Also known as dynamic predecessor search data structure.

a given step of the algorithm, and when we say current cost, current centers and current clusters we refer to the state of these objects at the end of the last local-search step before the current one. Let $O_1^* \ldots O_k^*$ be an optimal clustering of $P$ and let $\mathcal{O}^* = \{o_i = \mu(O_i^*) \mid \text{for } i = 1 \ldots k\}$ be the set of optimal centers of these clusters. We denote with $C_1 \ldots C_k$ the current set of clusters at that stage of the local search and with $\mathcal{C} = \{c_1 \ldots c_k\}$ the set of their respective current centers.

We say that $c_i$ *captures* $o_j$ if $c_i$ is the closest current center to $o_j$, namely $c_i = \mathcal{C}[o_j]$. We say that $c_i$ is *busy* if it captures more than $p$ optimal centers, and we say it is *lonely* if it captures no optimal center. Let $\widetilde{\mathcal{O}} = \{o_i \mid \text{cost}(O_i^*, \mathcal{C}) > \delta \cdot \text{ALG}/k\}$ and $\widetilde{\mathcal{C}} = \mathcal{C} \setminus \{\mathcal{C}[o_i] \mid o_i \in \mathcal{O}^* \setminus \widetilde{\mathcal{O}}\}$. For ease of notation, we simply assume that $\widetilde{\mathcal{O}} = \{o_1 \ldots o_h\}$ and $\widetilde{\mathcal{C}} = \{c_1 \ldots c_{h'}\}$. Notice that $h' > h$.

**Weighted ideal multi-swaps.** Given $In \subseteq P$ and $Out \subseteq \widetilde{\mathcal{C}}$ of the same size we say that the swap $(In, Out)$ is an *ideal* swap if $In \subseteq \widetilde{\mathcal{O}}$. We now build a set of *weighted* ideal multi-swaps $\mathcal{S}$. First, suppose wlog that $\{c_1 \ldots c_t\}$ is the set of current centers in $\widetilde{\mathcal{C}}$ that are neither lonely nor busy. Let $\mathcal{L}$ be the set of lonely centers in $\widetilde{\mathcal{C}}$. For each $i = 1 \ldots t$, we do the following. Let $In$ be the set of optimal centers in $\widetilde{\mathcal{O}}$ captured by $c_i$. Choose a set $\mathcal{L}_i$ of $|In| - 1$ centers from $\mathcal{L}$, set $\mathcal{L} \leftarrow \mathcal{L} \setminus \mathcal{L}_i$ and define $Out = \mathcal{L}_i \cup \{c_i\}$. Assign weight 1 to $(In, Out)$ and add it to $\mathcal{S}$. For each busy center $c_i \in \{c_{t+1} \ldots c_{h'}\}$ let $A$ be the set of optimal centers in $\widetilde{\mathcal{O}}$ captured by $c_i$, pick a set $\mathcal{L}_i$ of $|A| - 1$ lonely current centers from $\mathcal{L}$ (a counting argument shows that this is always possible). Set $\mathcal{L} \leftarrow \mathcal{L} \setminus \mathcal{L}_i$. For each $o_j \in A$ and $c_\ell \in \mathcal{L}_i$ assign weight $1/(|A| - 1)$ to $(o_j, c_\ell)$ and add it to $\mathcal{S}$. Suppose we are left with $\ell$ centers $o_1' \ldots o_\ell' \in \widetilde{\mathcal{O}}$ such that $\mathcal{C}[o_i'] \notin \widetilde{\mathcal{C}}$. Apparently, we have not included any $o_i'$ in any swap yet. However, since $|\widetilde{\mathcal{C}}| \geq |\widetilde{\mathcal{O}}|$, we are left with at least $\ell' \geq \ell$ lonely centers $c_1' \ldots c_{\ell'}' \in \widetilde{\mathcal{C}}$. For each $i = 1 \ldots \ell$ we assign weight 1 to $(o_i', c_i')$ and add it to $\mathcal{S}$.

**Observation 7.** *The process above generates a set of weighted ideal multi-swaps such that: (i) Every swap has size at most $p$; (ii) The combined weights of swaps involving an optimal center $o_i \in \widetilde{\mathcal{O}}$ is 1; (iii) The combined weights of swaps involving a current center $c_i$ is at most $1 + 1/p$.*

Consider an ideal swap $(In, Out)$. Let $O_{In}^* = \bigcup_{o_i \in In} O_i^*$ and $C_{Out} = \bigcup_{c_j \in Out} C_j$. Define the reassignment cost $\text{Reassign}(In, Out)$ as the increase in cost of reassigning points in $C_{Out} \setminus O_{In}^*$ to centers in $\mathcal{C} \setminus Out$. Namely,

$$\text{Reassign}(In, Out) = \text{cost}(C_{Out} \setminus O_{In}^*, \mathcal{C} \setminus Out) - \text{cost}(C_{Out} \setminus O_{In}^*, \mathcal{C}).$$

We take the increase in cost of the following reassignment as an upper bound to the reassignment cost. For each $p \in C_{Out} \setminus O_{In}^*$ we consider its closest optimal center $\mathcal{O}^*[p]$ and reassign $p$ to the current center that is closest to $\mathcal{O}^*[p]$, namely $\mathcal{C}[\mathcal{O}^*[p]]$. In formulas, we have

$$\text{Reassign}(In, Out) \leq \sum_{p \in C_{Out} \setminus O_{In}^*} \text{cost}(p, \mathcal{C}[\mathcal{O}^*[p]]) - \text{cost}(p, \mathcal{C}[p])$$

$$\leq \sum_{p \in C_{Out}} \text{cost}(p, \mathcal{C}[\mathcal{O}^*[p]]) - \text{cost}(p, \mathcal{C}[p]).$$

Indeed, by the way we defined our ideal swaps we have $\mathcal{C}[\mathcal{O}^*[p]] \notin Out$ for each $p \notin O_{In}^*$ and this reassignment is valid. Notice that the right hand side in the equation above does not depend on $In$.

**Lemma 8.** $\sum_{p \in P} \text{cost}(p, \mathcal{C}[\mathcal{O}^*[p]]) \leq 2OPT + ALG + 2\sqrt{ALG}\sqrt{OPT}$.

*Proof.* Deferred to the supplementary material. $\square$

**Lemma 9.** *The combined weighted reassignment costs of all ideal multi-swaps in $\mathcal{S}$ is at most $(2 + 2/p) \cdot (OPT + \sqrt{ALG}\sqrt{OPT})$.*

*Proof.* Denote by $w(In, Out)$ the weight associated with the swap $(In, Out)$.

$$\sum_{(In,Out)\in\mathcal{S}} w(In, Out) \cdot \texttt{Reassign}(In, Out) \leq$$

$$\sum_{(In,Out)\in\mathcal{S}} w(In, Out) \cdot \sum_{p\in C_{Out}} \texttt{cost}(p, \mathcal{C}[\mathcal{O}^*[p]]) - \texttt{cost}(p, \mathcal{C}[p]) \leq$$

$$(1 + 1/p) \cdot \sum_{c_j\in\mathcal{C}} \sum_{p\in C_j} \texttt{cost}(p, \mathcal{C}[\mathcal{O}^*[p]]) - \texttt{cost}(p, \mathcal{C}[p]) \leq$$

$$(1 + 1/p) \cdot \left( \sum_{p\in P} \texttt{cost}(p, \mathcal{C}[\mathcal{O}^*[p]]) - \text{ALG} \right).$$

The second inequality uses $(iii)$ from Observation 7. Applying Lemma 8 completes the proof. $\square$

Recall the notions of radius and core of an optimal cluster introduced in Section 2. We say that a swap $(In, Out)$ is *strongly improving* if $\texttt{cost}(P, (\mathcal{C} \cup In) \setminus Out) \leq (1 - \delta/k) \cdot \texttt{cost}(P, \mathcal{C})$. Let $In = \{o_1 \ldots o_s\} \subseteq \widetilde{\mathcal{O}}$ and $Out = \{c_1 \ldots c_s\} \subseteq \widetilde{\mathcal{C}}$ we say that an ideal swap $(In, Out)$ is *good* if for every $q_1 \in \texttt{core}(o_1) \ldots q_s \in \texttt{core}(o_s)$ the swap $(\mathcal{Q}, Out)$ is strongly improving, where $\mathcal{Q} = \{q_1 \ldots q_s\}$. We call an ideal swap *bad* otherwise. We say that an optimal center $o_i \in \widetilde{\mathcal{O}}$ is good if that's the case for at least one of the ideal swaps it belongs to, otherwise we say that it is bad. Notice that each optimal center in $\widetilde{\mathcal{O}}$ is assigned to a swap in $\mathcal{S}$, so it is either good or bad. Denote with $G$ the union of cores of good optimal centers in $\widetilde{\mathcal{O}}$.

**Lemma 10.** *If an ideal swap $(In, Out)$ is bad, then we have*

$$\texttt{cost}(O^*_{In}, \mathcal{C}) \leq (2 + \delta)\texttt{cost}(O^*_{In}, \mathcal{O}^*) + \texttt{Reassign}(In, Out) + \delta\text{ALG}/k. \qquad (2)$$

*Proof.* Let $In = \{o_1 \ldots o_s\}$, $\mathcal{Q} = \{q_1 \ldots q_s\}$ such that $q_1 \in \texttt{core}(o_1) \ldots q_s \in \texttt{core}(o_s)$. Then, by Lemma 1 $\texttt{cost}(O^*_{In}, \mathcal{Q}) \leq (2 + \delta)\texttt{cost}(O^*_{In}, \mathcal{O}^*)$. Moreover, $\texttt{Reassign}(In, Out) = \texttt{cost}(P \setminus O^*_{In}, \mathcal{C} \setminus Out) - \texttt{cost}(P \setminus O^*_{In}, \mathcal{C})$ because points in $P \setminus C_{Out}$ are not affected by the swap. Therefore, $\texttt{cost}(P, (\mathcal{C} \cup \mathcal{Q}) \setminus Out) \leq (2 + \delta)\texttt{cost}(O^*_{In}, \mathcal{O}^*) + \texttt{Reassign}(In, Out) + \texttt{cost}(P \setminus O^*_{In}, \mathcal{C})$. Suppose by contradiction that Equation (2) does not hold, then

$$\texttt{cost}(P, \mathcal{C}) - \texttt{cost}(P, (\mathcal{C} \cup \mathcal{Q}) \setminus Out) =$$
$$\texttt{cost}(P \setminus O^*_{In}, \mathcal{C}) + \texttt{cost}(O^*_{In}, \mathcal{C}) - \texttt{cost}(P, (\mathcal{C} \cup \mathcal{Q}) \setminus Out) \geq \delta\text{ALG}/k.$$

Hence, $(\mathcal{Q}, Out)$ is strongly improving and this holds for any choice of $\mathcal{Q}$, contradiction. $\square$

**Lemma 11.** *If $ALG/OPT > \eta^2 + \delta$ then $\texttt{cost}(G, \mathcal{C}) = \Omega_\delta(\texttt{cost}(P, \mathcal{C}))$. Thus, if we $D^2$-sample $q$ we have $P[q \in G] = \Omega_\delta(1)$.*

*Proof.* First, we observe that the combined current cost of all optimal clusters in $\mathcal{O}^* \setminus \widetilde{\mathcal{O}}$ is at most $k \cdot \delta\text{ALG}/k = \delta\text{ALG}$. Now, we prove that the combined current cost of all $O^*_i$ such that $o_i$ is bad is $\leq (1 - 2\delta)\text{ALG}$. Suppose, by contradiction, that it is not the case, then we have:

$$(1 - 2\delta)\text{ALG} < \sum_{\text{Bad } o_i\in\widetilde{\mathcal{O}}} \texttt{cost}(O^*_i, \mathcal{C}) \leq \sum_{\text{Bad } (In,Out)\in\mathcal{S}} w(In, Out) \cdot \texttt{cost}(O^*_{In}, \mathcal{C}) \leq$$

$$\sum_{\text{Bad } (In,Out)} w(In, Out) \cdot ((2 + \delta)\texttt{cost}(O^*_{In}, \mathcal{O}^*) + \texttt{Reassign}(In, Out) + \delta\text{ALG}/k) \leq$$

$$(2 + \delta)\text{OPT} + (2 + 2/p)\text{OPT} + (2 + 2/p)\sqrt{\text{ALG}}\sqrt{\text{OPT}} + \delta\text{ALG}.$$

The second and last inequalities make use of Observation 7. The third inequality uses Lemma 10.

Setting $\eta^2 = \text{ALG}/\text{OPT}$ we obtain the inequality $\eta^2 - (2 + 2/p \pm O(\delta))\eta - (4 + 2/p \pm O(\delta)) \leq 0$. Hence, we obtain a contradiction in the previous argument as long as $\eta^2 - (2 + 2/p \pm O(\delta))\eta - (4 + 2/p \pm O(\delta)) > 0$. A contradiction there implies that at least an $\delta$-fraction of the current cost is due to points in $\bigcup_{\text{Good } o_i\in\widetilde{\mathcal{O}}} O^*_i$. We combine this with Lemma 2 and conclude that the total current cost of $G = \bigcup_{\text{Good } o_i\in\widetilde{\mathcal{O}}} \texttt{core}(O^*_i)$ is $\Omega_\delta(\texttt{cost}(P, \mathcal{C}))$. $\square$

Finally, we prove Lemma 6. Whenever $q_1 \in G$ we have that $q_1 \in \mathtt{core}(o_1)$ for some good $o_1$. Then, for some $s \leq p$ we can complete $o_1$ with $o_2 \ldots o_s$ such that $In = \{o_1 \ldots o_s\}$ belongs to a good swap. Concretely, there exists $Out \subseteq \mathcal{C}$ such that $(In, Out)$ is a good swap. Since $In \subset \widetilde{\mathcal{O}}$ we have $\mathtt{cost}(O_i^*, \mathcal{C}) > \delta \mathrm{OPT}/k$ for all $o_i \in In$, which combined with Lemma 2 gives that for $i = 2 \ldots s$ $P[q_i \in \mathtt{core}(o_i)] \geq \Omega_\delta(1/k)$. Hence, we have $P[q_i \in \mathtt{core}(o_i)$ for $i = 1 \ldots s] \geq \Omega_{\delta,p}(1/k^{p-1})$. Whenever we sample $q_1 \ldots q_s$ from $\mathtt{core}(o_1) \ldots \mathtt{core}(o_s)$, we have that $(\mathcal{Q}, Out)$ is strongly improving. Notice, however, that $(\mathcal{Q}, Out)$ is a $s$-swap and we may have $s < p$. Nevertheless, whenever we sample $q_1 \ldots q_s$ followed by any sequence $q_{s+1} \ldots q_p$ it is enough to choose $Out' = Out \cup \{q_{s+1} \ldots q_p\}$ to obtain that $(\{q_1 \ldots q_p\}, Out')$ is an improving $p$-swap.

# 4 A Faster $(9 + \varepsilon)$-Approximation Local Search Algorithm

The MSLS algorithm from Section 3 achieves an approximation ratio of $\eta^2 + \varepsilon$, where $\eta^2 - (2 + 2/p)\eta - (4 + 2/p) = 0$ and $\varepsilon > 0$ is an arbitrary small constant. For large $p$ we have $\eta \approx 10.48$. On the other hand, employing $p$ simultaneous swaps, Kanungo et al. [2004] achieve an approximation factor of $\xi^2 + \varepsilon$ where $\xi^2 - (2 + 2/p)\xi - (3 + 2/p) = 0$. If we set $p \approx 1/\varepsilon$ this yields a $(9 + O(\varepsilon))$-approximation. In the same paper, they prove that 9-approximation is indeed the best possible for $p$-swap local search, if $p$ is constant (see Theorem 3.1 in Kanungo et al. [2004]). They showed that 9 is the right locality gap for local search, but they matched it with a very slow algorithm. To achieve a $(9 + \varepsilon)$-approximation, they discretize the space reducing to $O(n\varepsilon^{-d})$ candidate centers and perform an exhaustive search over all size-$(1/\varepsilon)$ subsets of candidates at every step. As we saw in the related work section, it is possible to combine techniques from coreset and dimensionality reduction to reduce the number of points to $n' = k \cdot poly(\varepsilon^{-1})$ and the number of dimensions to $d' = \log k \cdot \varepsilon^{-2}$. This reduces the complexity of Kanungo et al. [2004] to $k^{O(\varepsilon^{-3} \log \varepsilon^{-1})}$.

In this section, we leverage techniques from Cohen-Addad et al. [2021] to achieve a $(9 + \varepsilon)$-approximation faster [4]. In particular, we obtain the following.

**Theorem 12.** *Given a set of $n$ points in $\mathbb{R}^d$ with aspect ratio $\Delta$, there exists an algorithm that computes a $9 + \varepsilon$-approximation to $k$-means in time $ndk^{O(\varepsilon^{-2})} \log^{O(\varepsilon^{-1})}(\Delta) \cdot 2^{-poly(\varepsilon^{-1})}$.*

Notice that, besides being asymptotically slower, the pipeline obtained combining known techniques is highly impractical and thus it did not make for an experimental test-bed. Moreover, it is not obvious how to simplify such an ensemble of complex techniques to obtain a practical algorithm.

**Limitations of MSLS.** The barrier we need to overcome in order to match the bound in Kanungo et al. [2004] is that, while we only consider points in $P$ as candidate centers, the discretization they employ considers also points in $\mathbb{R}^d \setminus P$. In the analysis of MSLS we show that we sample each point $q_i$ from $\mathtt{core}(O_i^*)$ or equivalently that $q_i \in B(o_i, (1 + \epsilon)\rho_i)$, where $\rho_i$ is such that $O_i^*$ would have the same cost w.r.t. $o_i$ if all its points were moved on a sphere of radius $\rho_i$ centered in $o_i$. This allows us to use a Markov's inequality kind of argument and conclude that there must be $\Omega_\epsilon(|O_i^*|)$ points in $O_i^* \cap B(o_i, (1 + \epsilon)\rho_i)$. However, we have no guarantee that there is any point at all in $O_i^* \cap B(o_i, (1 - \varepsilon)\rho_i)$. Indeed, all points in $O_i^*$ might lie on $\partial B(o_i, \rho_i)$. The fact that potentially all our candidate centers $q$ are at distance at least $\rho_i$ from $o_i$ yields (by Lemma 1) $\mathtt{cost}(O_i^*, q) \geq 2\mathtt{cost}(O_i^*, o_i)$, which causes the zero-degree term in $\xi^2 - (2+2/p)\xi - (3+2/p) = 0$ from Kanungo et al. [2004] to become a 4 in our analysis.

**Improving MSLS by taking averages.** First, we notice that, in order to achieve $(9 + \varepsilon)$-approximation we need to set $p = \Theta(1/\varepsilon)$. The main hurdle to achieve a $(9 + \varepsilon)$-approximation is that we need to replace the $q_i$ in MSLS with a better approximation of $o_i$. We design a subroutine that computes, with constant probability, an $\varepsilon$-approximation $\hat{o}_i$ of $o_i$ (namely, $\mathtt{cost}(O_i^*, \hat{o}_i) \leq (1 + \varepsilon)\mathtt{cost}(O_i^*, o_i)$). The key idea is that, if sample uniformly $O(1/\varepsilon)$ points from $O_i^*$ and define $\hat{o}_i$ to be the average of our samples then $\mathtt{cost}(O_i^*, \hat{o}_i) \leq (1 + \varepsilon)\mathtt{cost}(O_i^*, o_i)$

Though, we do not know $O_i^*$, so sampling uniformly from it is non-trivial. To achieve that, for each $q_i$ we identify a set $N$ of *nice* candidate points in $P$ such that a $poly(\varepsilon)/k$ fraction of them are from $O_i^*$. We sample $O(1/\varepsilon)$ points uniformly from $N$ and thus with probability $(\varepsilon/k)^{O(1/\varepsilon)}$ we sample only points from $O_i^*$. Thus far, we sampled $O(1/\varepsilon)$ points uniformly from $N \cap O_i^*$. What about

---

[4]The complexity in Theorem 12 can be improved by applying the same preprocessing techniques using coresets and dimensionality reduction, similar to what can be used to speed up the approach of Kanungo et al. [2004]. Our algorithm hence becomes asymptotically faster.

the points in $O_i^* \setminus N$? We can define $N$ so that all points in $O_i^* \setminus N$ are either very close to some of the $(q_j)_j$ or they are very far from $q_i$. The points that are very close to points $(q_j)_j$ are easy to treat. Indeed, we can approximately locate them and we just need to guess their mass, which is matters only when $\geq poly(\varepsilon)$ALG, and so we pay only a $\log^{O(1/\varepsilon)}(1/\varepsilon)$ multiplicative overhead to guess the mass close to $q_j$ for $j = 1 \ldots p = \Theta(1/\varepsilon)$. As for a point $f$ that is very far from $q_i$ (say, $||f - q_i|| \gg \rho_i$) we notice that, although $f$'s contribution to $\mathtt{cost}(O_i^*, o_i)$ may be large, we have $\mathtt{cost}(f, o) \approx \mathtt{cost}(f, o_i)$ for each $o \in B(q_i, \rho_i) \subseteq B(o_i, (2 + \varepsilon)\rho_i)$ assuming $q_i \in \mathtt{core}(o_i)$.

# 5 Experiments

In this section, we show that our new algorithm using multi-swap local search can be employed to design an efficient seeding algorithm for Lloyd's which outperforms both the classical $k$-means++ seeding and the single-swap local search from Lattanzi and Sohler [2019].

**Algorithms.** The multi-swap local search algorithm that we analysed above performs very well in terms of solution quality. This empirically verifies the improved approximation factor of our algorithm, compared to the single-swap local search of Lattanzi and Sohler [2019].

Motivated by practical considerations, we heuristically adapt our algorithm to make it very competitive with SSLS in terms of running time and still remain very close, in terms of solution quality, to the theoretically superior algorithm that we analyzed. The adaptation of our algorithm replaces the phase where it selects the $p$ centers to swap-out by performing an exhaustive search over $\binom{k+p}{p}$ subsets of centers. Instead, we use an efficient heuristic procedure for selecting the $p$ centers to swap-out, by greedily selecting one by one the centers to swap-out. Specifically, we select the first center to be the cheapest one to remove (namely, the one that increases the cost by the least amount once the points in its cluster are reassigned to the remaining centers). Then, we update all costs and select the next center iteratively. After $p$ repetitions we are done. We perform an experimental evaluation of the "greedy" variant of our algorithm compared to the theoretically-sound algorithm from Section 3 and show that employing the greedy heuristic does not measurably impact performance.

The four algorithms that we evaluate are the following: 1) **KM++:** The $k$-means++ from Arthur and Vassilvitskii [2007], 2) **SSLS:** The Single-swap local search method from Lattanzi and Sohler [2019], 3) **MSLS:** The multi-swap local search from Section 3, and 4) **MSLS-G:** The greedy variant of multi-swap local search as described above.

We use MSLS-G-$p = x$ and MSLS-$p = x$, to denote MSLS-G and MSLS with $p = x$, respectively. Notice that MSLS-G-$p = 1$ is exactly SSLS. Our experimental evaluation explores the effect of $p$-swap LS, for $p > 1$, in terms of solution cost and running time.

**Datasets.** We consider the three datasets used in Lattanzi and Sohler [2019] to evaluate the performance of SSLS: 1) KDD-PHY – $100,000$ points with $78$ features representing a quantum physic task kdd [2004], 2) RNA - $488,565$ points with $8$ features representing RNA input sequence pairs Uzilov et al. [2006], and 3) KDD-BIO – $145,751$ points with $74$ features measuring the match between a protein and a native sequence kdd [2004]. We discuss the results for two or our datasets, namely KDD-BIO and RNA. We deffer the results on KDD-PHY to the appendix and note that the results are very similar to the results on RNA.

We performed a preprocessing step to clean-up the datasets. We observed that the standard deviation of some features was disproportionately high. This causes all costs being concentrated in few dimensions making the problem, in some sense, lower-dimensional. Thus, we apply min-max scaling to all datasets and observed that this causes all our features' standard deviations to be comparable.

**Experimental setting.** All our code is written in Python. The code will be made available upon publication of this work. We did not make use of parallelization techniques. To run our experiments, we used a personal computer with $8$ cores, a $1.8$ Ghz processor, and $15.9$ GiB of main memory We run all experiments $5$ times and report the mean and standard deviation in our plots. All our plots report the progression of the cost either w.r.t local search steps, or Lloyd's iterations. We run experiments on all our datasets for $k = 10, 25, 50$. The main body of the paper reports the results for $k = 25$, while the rest can be found in the appendix. We note that the conclusions of the experiments for $k = 10, 50$ are similar to those of $k = 25$.

**Removing centers greedily.** We first we compare MSLS-G with MSLS. To perform our experiment, we initialize $k = 25$ centers using $k$-means++ and then run $50$ iterations of local search for both

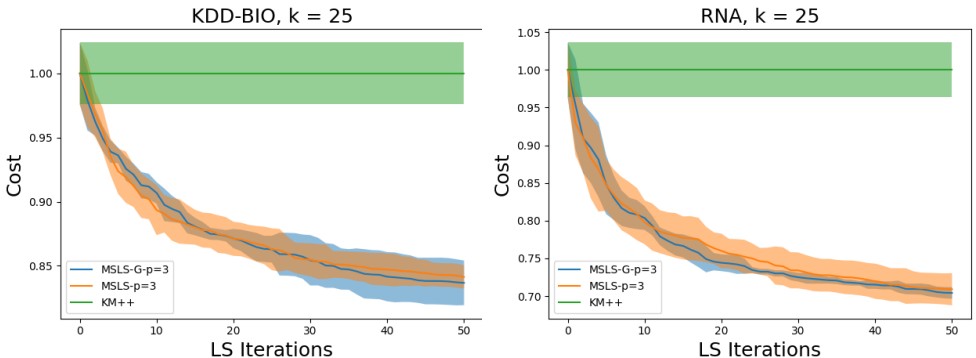

Figure 1: Comparison between MSLS and MSLS-G, for $p = 3$, for $k = 25$, on the datasets KDD-BIO and RNA. The $y$ axis shows the solution cost divided by the means solution cost of KM++.

algorithms, for $p = 3$ swaps. Due to the higher running of the MSLS we perform this experiments on 1% uniform sample of each of our datasets. We find out that the performance of the two algorithms is comparable on all our instances, while they both perform roughly 15%-27% at convergence. Figure 1 shows the aggregate results, over 5 repetitions of our experiment.

It may happen that MSLS, which considers all possible swaps of size $p$ at each LS iteration, performs worse than MSLS-G as a sub-optimal swap at intermediate iterations may still lead to a better local optimum by coincidence. Given that MSLS-G performs very comparably to MSLS, while it is much faster in practice, we use MSLS-G for the rest of our experiments where we compare to baselines. This allows us to consider higher values of $p$, without compromising much the running time.

**Results: Evaluating the quality and performance of the algorithms.** In our first experiment we run KM++ followed by 50 iterations of MSLS-G with $p = 1, 4, 7, 10$ and plot the relative cost w.r.t. KM++ at each iteration, for $k = 25$. The first row of Figure 2 plots the results. Our experiment shows that, after 50 iterations MSLS-G for $p = 4, 7, 10$ achieves improvements of roughly 10% compared to MSLS-G-$p = 1$ and of the order of $20\% - 30\%$ compared to KM++. We also report the time per iteration that each algorithm takes. For comparison, we report the running time of a single iteration of Lloyd's next to the dataset's name. It is important to notice that, although MSLS-G-$p = 1$ is faster, running more iterations MSLS-G-$p = 1$ is not sufficient to compete with MSLS-G when $p > 1$.

**Results: Evaluating the quality after postprocessing using Lloyd.** In our second experiment, we use KM++ and MSLS-G as a seeding algorithm for Lloyd's and measure how much of the performance improvement measured in the first experiment is retained after running Lloyd's. First, we initialize our centers using KM++ and the run 15 iterations of MSLS-G for $p = 1, 4, 7$. We measure the cost achieved by running 10 iterations of Lloyd's starting from the solutions found by MSLS-G as well as KM++. In Figure 2 (second row) we plot the results. Notice that, according to the running times from the first experiment, 15 iterations iterations of MSLS-G take less than 10 iterations of Lloyd's for $p = 4, 7$ (and also for $p = 10$, except on RNA). We observe that MSLS-G for $p > 1$ performs at least as good as SSLS from Lattanzi and Sohler [2019] and in some cases maintains non-trivial improvements.

**Results: Evaluating the quality and performance of the algorithms against a fixed deadline.** In this experiment we run KM++ followed by MSLS-G with $p = 1, 4, 7, 10$, for a set of fixed amounts of time. This setting allows the versions of MSLS-G with smaller swap size to perform more iterations compared to the versions of the algorithm with a larger swap size, as smaller swap size leads to lower running time per iteration. Let $\tau$ be the average time that Lloyd's algorithm requires to complete a simple iteration on a specific instance. We plot the cost of the solution produced by each algorithm after running $\lambda \times \tau$ for each $\lambda \in \{1, \cdots, 20\}$ in Figure 3. Our experiment shows that MSLS-G for $p = 4, 7, 10$ achieves improvements of more than 5% compared to MSLS-G-$p = 1$ even when compared against a fixed running time, and of the order of $20\% - 30\%$ compared to KM++.

## Conclusion and Future Directions

We present a new algorithm for the $k$-means problem and we show that it outperforms theoretically and experimentally state-of-the-art practical algorithms with provable guarantees in terms of solution

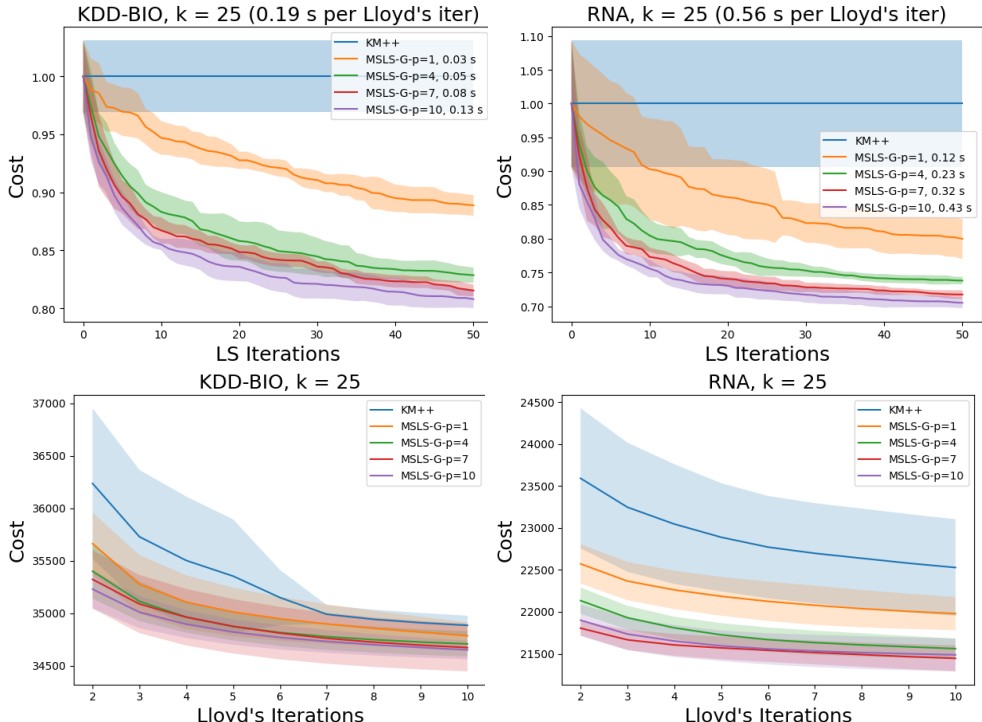

Figure 2: The first row compares the cost of MSLS-G, for $p \in \{1, 4, 7, 10\}$, divided by the mean cost of KM++ at each LS step, for $k = 25$. The legend reports also the running time of MSLS-G per LS step (in seconds). The second row compares the cost after each of the 10 iterations of Lloyd with seeding from MSLS-G, for $p \in \{1, 4, 7, 10\}$ and 15 local search steps and KM++, for $k = 25$.

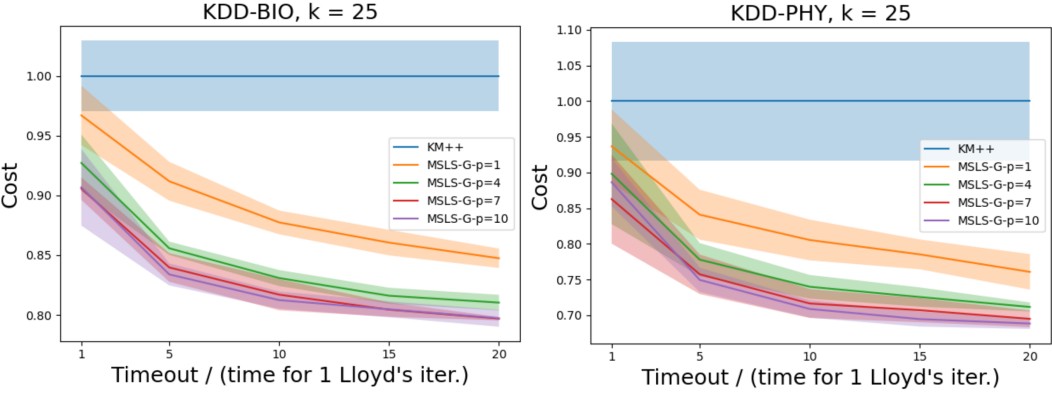

Figure 3: Comparison of the cost produced by MSLS-G, for $p \in \{1, 4, 7, 10\}$ and $k = 25$ on the datasets KDD-BIO and KDD-PHU, divided by the mean cost of KM++ after running for fixed amount of time in terms of multiplicative factors to the average time for an iteration of Lloyd's algorithm (i.e., for deadlines that are $1\times, \ldots, 20\times$ the average time of an iteration of Lloyd).

quality. A very interesting open question is to improve our local search procedure by avoiding the exhaustive search over all possible size-$p$ subsets of centers to swap out, concretely an algorithm with running time $\tilde{O}(2^{poly(1/\varepsilon)}ndk)$.

**Acknowledgements.** This work was partially done when Lorenzo Beretta was a Research Student at Google Research. Moreover, Lorenzo Beretta receives funding from the European Union's Horizon 2020 research and innovation program under the Marie Skłodowska-Curie grant agreement No. 801199.

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

# Supplementary Material

## Proofs from Section 3

**Lemma 8.** $\sum_{p \in P} cost(p, \mathcal{C}[\mathcal{O}^*[p]]) \leq 2OPT + ALG + 2\sqrt{ALG}\sqrt{OPT}.$

*Proof.*

$$\sum_{p \in P} \texttt{cost}(p, \mathcal{C}[\mathcal{O}^*[p]]) =$$

$$\sum_{o_i \in \mathcal{O}^*} \sum_{p \in O_i^*} \texttt{cost}(p, \mathcal{C}[o_i]) =$$

$$\sum_{o_i \in \mathcal{O}^*} |O_i^*| \cdot \texttt{cost}(o_i, \mathcal{C}[o_i]) + \texttt{cost}(O_i^*, o_i) =$$

$$\text{OPT} + \sum_{p \in P} \texttt{cost}(\mathcal{O}^*[p], \mathcal{C}[\mathcal{O}^*[p]]) \leq$$

$$\text{OPT} + \sum_{p \in P} \texttt{cost}(\mathcal{O}^*[p], \mathcal{C}[p]) \leq$$

$$\text{OPT} + \sum_{p \in P} \left( ||\mathcal{O}^*[p] - p|| + ||p - \mathcal{C}[p]|| \right)^2 =$$

$$2\text{OPT} + \text{ALG} + 2\sum_{p \in P} ||\mathcal{O}^*[p], p|| \cdot ||p, \mathcal{C}[p]|| \leq 2\text{OPT} + \text{ALG} + 2\sqrt{\text{ALG}}\sqrt{\text{OPT}}.$$

The second equality is due to Lemma 1 and the last inequality is due to Cauchy-Schwarz. $\qquad\square$

## Proofs from Section 4

In this section, we prove the following.

**Theorem 12.** *Given a set of $n$ points in $\mathbb{R}^d$ with aspect ratio $\Delta$, there exists an algorithm that computes a $9 + \varepsilon$-approximation to $k$-means in time $ndk^{O(\varepsilon^{-2})} \log^{O(\varepsilon^{-1})}(\Delta) \cdot 2^{-poly(\varepsilon^{-1})}$.*

We start with a key lemma showing that a sample of size $O(1/\varepsilon)$ is enough to approximate 1-mean.

**Lemma 13** (Form Inaba et al. [1994]). *Given an instance $P \subseteq \mathbb{R}^d$, sample $m = 1/(\varepsilon\delta)$ points uniformly at random from $P$ and denote the set of samples with $S$. Then $\texttt{cost}(P, \mu(S)) \leq (1 + \varepsilon)\texttt{cost}(P, \mu(P))$ with probability at least $1 - \delta$.*

*Proof.* We want to prove that with probability $1 - \delta$ we have $||\mu(S) - \mu(P)||^2 \leq \varepsilon\texttt{cost}(P, \mu(P))/|P|$. Then, applying Lemma 1 gives the desired result. First, we notice that $\mu(P)$ is an unbiased estimator of $\mu(P)$, namely $E[\mu(S)] = \mu(P)$. Then, we have

$$E\left[||\mu(S) - \mu(P)||^2\right] = \frac{1}{m} \sum_{i=1}^{|S|} E\left[||s_i - \mu(P)||^2\right] = \frac{\texttt{cost}(P, \mu(P))}{m \cdot |P|}$$

where $s_i$ are uniform independent samples from $P$. Applying Markov's inequality concludes the proof. $\qquad\square$

The algorithm that verifies Theorem 12 is very similar to the MSLS algorithm from Section 3 and we use the same notation to describe it. The intuition is that in MSLS we sample $\mathcal{Q} = \{q_1 \ldots q_p\}$ hoping that $q_i \in \texttt{core}(o_i)$ for each $i$; here we refine $q_i$ to a better approximation $\hat{o}_i$ of $o_i$ and swap the points $(\hat{o}_i)_i$ rather than $(q_i)_i$. Our points $\hat{o}_i$ are generated taking the average of some sampled point, thus we possibly have $\hat{o}_i \notin P$ while, on the other hand, $q_i \in P$.

**A $(9+\varepsilon)$-approximation MSLS algortihm.** First, we initialize our set of centers using $k$-means++. Then, we run $ndk^{O(\varepsilon^{-2})} \cdot 2^{poly(\varepsilon^{-1})}$ local search steps, where a local search step works as follows. Set $p = \Theta(\varepsilon^{-1})$. We $D^2$-sample a set $\mathcal{Q} = \{q_1 \ldots q_p\}$ of points from $P$ (without updating costs). Then, we iterate over all possible sets $Out = \{c_1 \ldots c_p\}$ of $p$ distinct elements in $\mathcal{C} \cup \mathcal{Q}$. We define the set of *temporary* centers $\mathcal{T} = (\mathcal{C} \cup \mathcal{Q}) \setminus Out$ and run a subroutine APX-CENTERS($\mathcal{T}$) which returns a list of $poly(\varepsilon^{-1}) \cdot \log^{O(\varepsilon^{-1})}(\Delta)$ size-$s$ sets $\widehat{In} = \{\hat{o}_1 \ldots \hat{o}_s\}$ (where $s = |\mathcal{Q} \setminus Out|$). We select the set $\widehat{In}$ in this list such that the swap $(\widehat{In}, Out \setminus \mathcal{Q})$ yields the maximum cost reduction. Then we select the set $Out$ that maximizes the cost reduction obtained in this way. If $(\widehat{In}, Out \setminus \mathcal{Q})$ actually reduces the cost then we perform that swap.

**A subroutine to approximate optimal centers.** Here we describe the subroutine APX-CENTERS($\mathcal{T}$). Let $\mathcal{Q} \setminus Out = \{q_1 \ldots q_s\}$. Recall that $s \le p = O(\varepsilon^{-1})$. This subroutine outputs a list of $2^{poly(\varepsilon^{-1})} \cdot \log^{O(\varepsilon^{-1})}(\Delta)$ size-$s$ sets $\widehat{In} = \{\hat{o}_1 \ldots \hat{o}_s\}$. Here we describe how to find a list of $2^{poly(\varepsilon^{-1})} \cdot \log(\Delta)$ values for $\hat{o}_1$. The same will apply for $\hat{o}_2 \ldots \hat{o}_s$ and taking the Cartesian product yields a list of $2^{poly(\varepsilon^{-1})} \cdot \log^{O(\varepsilon^{-1})}(\Delta)$ size-$s$ sets. Assume wlog that the pairwise distances between points in $P$ lie in $[1, \Delta]$. We iterate over all possible values of $\rho_1 \in \{1, (1+\varepsilon) \ldots (1+\varepsilon)^{\lceil \log_{1+\varepsilon} \Delta \rceil}\}$. We partition $P$ in three sets: the set of *far points* $F = \{x \in P \,|\, cost(x, q_1) > \rho_1^2/\varepsilon^3\}$, the set of *close points* $C = \{x \in P \setminus F \,|\, cost(x, \mathcal{T}) \le \varepsilon^3 \rho_1^2\}$ and the set of *nice points* $N = P \setminus (C \cup F)$. Then, we sample uniformly from $N$ a set $S$ of size $\Theta(\varepsilon^{-1})$. For each $(s+1)$-tuple of coefficients $\alpha_0, \alpha_1 \ldots \alpha_s \in \left\{1, (1-\varepsilon), (1-\varepsilon)^2, \ldots (1-\varepsilon)^{\lceil \log_{1-\varepsilon}(\varepsilon^7) \rceil}\right\} \cup \{0\}$ we output the candidate solution given by the convex combination

$$\hat{o}_1 = \hat{o}_1(\alpha_0 \ldots \alpha_s) = \frac{\alpha_0 \mu(S) + \sum_{i=1}^{s} \alpha_i q_i}{\sum_{i=0}^{s} \alpha_i} \tag{3}$$

so, for each value of $\rho_1$, we output $2^{poly(\varepsilon^{-1})}$ values for $\hat{o}_1$. Hence, $2^{poly(\varepsilon^{-1})} \cdot \log(\Delta)$ values in total.

**Analysis**

The key insight in the analysis of the MSLS algorithm form Section 3 was that every $q_i$ was a proxy for $o_i$ because $q_i \in \texttt{core}(o_i)$, and thus $q_i$ provided a good center for $O_i^*$. In the analysis of this improved version of MSLS we replace $q_i$ with $\hat{o}_i$ which makes a better center for $O_i^*$. Formally, fixed $Out$, we say that a point $\hat{o}_i$ is a *perfect approximation* of $o_i$ when $\texttt{cost}(O_i^*, (\mathcal{C} \cup \{\hat{o}_i\}) \setminus Out) \le (1+\varepsilon)\text{OPT}_i + \varepsilon\text{OPT}/k$. We define $\widetilde{\mathcal{O}}$ and $\widetilde{\mathcal{C}}$ as in Section 3, except that we replace $\delta$ with $\varepsilon$ (which here is not assumed to be a constant). Likewise, we build the set $\mathcal{S}$ of ideal multi-swaps as in Section 3. Recall that we say that a multi-swap $(In, Out)$ is *strongly improving* if $\texttt{cost}(P, (\mathcal{C} \cup In) \setminus Out) \le (1 - \varepsilon/k) \cdot \texttt{cost}(P, \mathcal{C})$. Let $In = \{o_1 \ldots o_s\} \subseteq \widetilde{\mathcal{O}}$ and $Out = \{c_1 \ldots c_s\} \subseteq \widetilde{\mathcal{C}}$, we overload the definition from Section 3 and say that the ideal multi-swap $(In, Out)$ is *good* if for every $\widehat{In} = \{\hat{o}_1 \ldots \hat{o}_s\}$ such that each $\hat{o}_i$ is a perfect approximation of $o_i$ for each $i = 1 \ldots s$ the swap $(\widehat{In}, Out)$ is strongly improving. We call an ideal swap *bad* otherwise. As in Section 3, we define the *core* of an optimal center; once again we replace $\delta$ with $\epsilon$, which is no longer constant. The two following lemmas are our stepping stones towards Theorem 12.

**Lemma 14.** *If $ALG/OPT > 9 + O(\varepsilon)$ then, with probability $k^{-O(\varepsilon^{-1})} \cdot 2^{-poly(\varepsilon^{-1})}$, there exists $Out \subseteq \mathcal{C} \cup \mathcal{Q}$ such that:*

*(i) If $\mathcal{Q} \setminus Out = \{q_1 \ldots q_s\}$ then $q_1 \in \texttt{core}(o_1) \ldots q_s \in \texttt{core}(o_s)$ for some $o_1 \ldots o_s \in \mathcal{O}^*$*

*(ii) If we define $In = \{o_1 \ldots o_s\}$ then $(In, Out \setminus \mathcal{Q})$ is a good ideal swap.*

**Lemma 15.** *If $(i)$ from Lemma 14 holds, then with probability $k^{-O(\varepsilon^{-2})} \cdot 2^{-poly(\varepsilon^{-1})}$, the list returned by APX-CENTERS contains $\widehat{In} = \{\hat{o}_1 \ldots \hat{o}_s\}$ such that $\hat{o}_i$ is a perfect approximation of $o_i$ for each $i = 1 \ldots s$.*

*Proof of Theorem 12.* Here we prove that our improved MSLS algorithm achieves a $(9 + O(\varepsilon))$-approximation, which is equivalent to Theorem 12 up to rescaling $\varepsilon$. Combining Lemma 14 and

Lemma 15 we obtain that, as long as ALG/OPT $> 9 + O(\varepsilon)$, with probability at least $k^{-O(\varepsilon^{-2})} \cdot 2^{-poly(\varepsilon^{-1})}$, the list returned by APX-CENTERS contains $\widehat{In} = \{\hat{o}_1 \ldots \hat{o}_s\}$ such that $(\widehat{In}, Out \setminus \mathcal{Q})$ is strongly improving. If this happens, we call such a local step *successful*. Now the proof goes exactly as the proof of Theorem 3. Indeed, We show that $k^{O(\varepsilon^{-2})} \cdot 2^{poly(\varepsilon^{-1})}$ local steps suffice to obtain $\Omega(k \log \log k/\varepsilon)$ successful local steps, and thus to obtain the desired approximation ratio, with constant probability.

To prove the running time bound it is sufficient to notice that a local search step can be performed in time $nd \log^{O(\varepsilon^{-1})}(\Delta) \cdot 2^{poly(\varepsilon^{-1})}$. $\qquad\square$

In the rest of this section, we prove Lemma 14 and Lemma 15.

**Observation 16.** *If we assume $\delta = \varepsilon$ non-constant in Lemma 2, then performing the computations explicitly we obtain* $\Pr[q \in core(O_i^*)] \geq poly(\varepsilon)$.

In order to prove Lemma 14, we first prove the two lemmas. Lemma 17 is the analogous of Lemma 10 and Lemma 18 is the analogous of Lemma 11. Overloading once again the definition from Section 3, we define $G$ as the union of cores of good optimal centers in $\widetilde{\mathcal{O}}$, where an optimal center is defined to be good if at least one of the ideal multi-swaps in $\mathcal{S}$ it belongs to is good (exactly as in Section 3).

**Lemma 17.** *If an ideal swap $(In, Out)$ is bad, then we have*

$$cost(O_{In}^*, \mathcal{C}) \leq (1 + \varepsilon) cost(O_{In}^*, \mathcal{O}^*) + \texttt{Reassign}(In, Out) + \varepsilon ALG/k. \tag{4}$$

*Proof.* Let $In = \{o_1 \ldots o_s\}$, $\widehat{In} = \{\hat{o}_1 \ldots \hat{o}_s\}$ such that $\hat{o}_i$ is a perfect approximation of $o_i$ for each $i = 1 \ldots s$. Recall that $O_{In}^* := \bigcup_{i=1}^s O_i^*$, then

$$\texttt{cost}\Big(O_{In}^*, (\mathcal{C} \cup \widehat{In}) \setminus Out\Big) \leq \sum_{i=1}^s \texttt{cost}(O_i^*, (\mathcal{C} \cup \{\hat{o}_i\}) \setminus Out) \leq (1 + \varepsilon)\texttt{cost}(O_{In}^*, \mathcal{O}^*). \tag{5}$$

Moreover, $\texttt{Reassign}(In, Out) = \texttt{cost}(P \setminus O_{In}^*, \mathcal{C} \setminus Out) - \texttt{cost}(P \setminus O_{In}^*, \mathcal{C})$ because points in $P \setminus C_{Out}$ are not affected by the swap. Therefore, $\texttt{cost}\Big(P, (\mathcal{C} \cup \widehat{In}) \setminus Out\Big) \leq (1 + \varepsilon)\texttt{cost}(O_{In}^*, \mathcal{O}^*) + \texttt{Reassign}(In, Out) + \texttt{cost}(P \setminus O_{In}^*, \mathcal{C})$. Suppose by contradiction that Equation (4) does not hold, then

$$\texttt{cost}(P, \mathcal{C}) - \texttt{cost}\Big(P, (\mathcal{C} \cup \widehat{In}) \setminus Out\Big) =$$

$$\texttt{cost}(P \setminus O_{In}^*, \mathcal{C}) + \texttt{cost}(O_{In}^*, \mathcal{C}) - \texttt{cost}\Big(P, (\mathcal{C} \cup \widehat{In}) \setminus Out\Big) \geq \epsilon ALG/k.$$

Hence, $(\widehat{In}, Out)$ is strongly improving and this holds for any choice of $\widehat{In}$, contradiction. $\qquad\square$

**Lemma 18.** *If $ALG/OPT > 9 + O(\varepsilon)$ then $cost(G, \mathcal{C}) \geq cost(P, \mathcal{C}) \cdot poly(\varepsilon)$. Thus, if we $D^2$-sample $q$ we have $P[q \in G] \geq poly(\varepsilon)$.*

*Proof.* First, we observe that the combined current cost of all optimal clusters in $\mathcal{O}^* \setminus \widetilde{\mathcal{O}}$ is at most $k \cdot \varepsilon ALG/k = \varepsilon ALG$. Now, we prove that the combined current cost of all $O_i^*$ such that $o_i$ is bad is $\leq (1 - 2\varepsilon)ALG$. Suppose, by contradiction, that it is not the case, then we have:

$$(1 - 2\varepsilon)ALG < \sum_{\text{Bad } o_i \in \widetilde{\mathcal{O}}} \texttt{cost}(O_i^*, \mathcal{C}) \leq \sum_{\text{Bad } (In, Out) \in \mathcal{S}} w(In, Out) \cdot \texttt{cost}(O_{In}^*, \mathcal{C}) \leq$$

$$\sum_{\text{Bad } (In, Out)} w(In, Out) \cdot ((1 + \varepsilon)\texttt{cost}(O_{In}^*, \mathcal{O}^*) + \texttt{Reassign}(In, Out) + \varepsilon ALG/k) \leq$$

$$(1 + \varepsilon)OPT + (2 + 2/p)OPT + (2 + 2/p)\sqrt{ALG}\sqrt{OPT} + \varepsilon ALG.$$

The second and last inequalities make use of Observation 7. The third inequality uses Lemma 17.

Setting $\eta^2 = ALG/OPT$ we obtain the inequality $\eta^2 - (2 + 2/p \pm O(\varepsilon))\eta - (3 + 2/p \pm O(\varepsilon)) \leq 0$. Hence, we obtain a contradiction in the previous argument as long as $\eta^2 - (2 + 2/p \pm O(\varepsilon))\eta - (3 +$

$2/p \pm O(\varepsilon)) > 0$, which holds for $p = \Theta(\varepsilon^{-1})$ and $\eta^2 = 9 + O(\varepsilon)$. A contradiction there implies that at least an $\varepsilon$-fraction of the current cost is due to points in $\bigcup_{\text{Good } o_i \in \widetilde{\mathcal{O}}} O_i^*$. Thanks to Observation 16, we have $P_{q \sim \text{cost}(q,\mathcal{C})}[q \in \text{core}(O_i^*) \mid q \in O_i^*] \geq poly(\varepsilon)$. Therefore, we can conclude that the current cost of $G = \bigcup_{\text{Good } o_i \in \widetilde{\mathcal{O}}} \text{core}(O_i^*)$ is at least a $poly(\varepsilon)$-fraction of the total current cost. $\qquad\square$

*Proof of Lemma 14.* Thanks to Lemma 18, we have that $P[q_1 \in G] \geq poly(\varepsilon)$. Whenever $q_1 \in G$ we have that $q_1 \in \text{core}(o_1)$ for some good $o_1$. Then, for some $s \leq p$ we can complete $o_1$ with $o_2 \ldots o_s$ such that $In = \{o_1 \ldots o_s\}$ belongs to a good swap. Concretely, there exists $Out \subseteq \mathcal{C}$ such that $(In, Out)$ is a good swap. Since $In \subset \widetilde{\mathcal{O}}$ we have $\text{cost}(O_i^*, \mathcal{C}) > \varepsilon \text{OPT}/k$ for all $o_i \in In$, which combined with Observation 16 gives that, for each $i = 2 \ldots s$, $P[q_i \in \text{core}(o_i)] \geq poly(\varepsilon)/k$. Hence, we have $P[q_i \in \text{core}(o_i) \text{ for } i = 1 \ldots s] \geq 2^{-poly(\varepsilon^{-1})} k^{-O(\varepsilon^{-1})}$. Notice, however, that $(\widehat{In}, Out)$ is a $s$-swap and we may have $s < p$. Nevertheless, whenever we sample $q_1 \ldots q_s$ followed by any sequence $q_{s+1} \ldots q_p$ it is enough to choose $Out' = Out \cup \{q_{s+1} \ldots q_p\}$ to obtain that $(\{q_1 \ldots q_p\}, Out')$ is an improving $p$-swap. $\qquad\square$

In order to prove Lemma 15 we first need a few technical lemmas.

**Lemma 19** (Lemma 2 from Lattanzi and Sohler [2019]). *For each $x, y, z \in \mathbb{R}^d$ and $\varepsilon > 0$, $\text{cost}(x, y) \leq (1 + \varepsilon)\text{cost}(x, z) + (1 + 1/\varepsilon)\text{cost}(z, y)$.*

**Lemma 20.** *Given $q \in \mathbb{R}^d$ and $Z \subseteq \mathbb{R}^d$ such that $\text{cost}(Z, q) \leq \varepsilon^2 \Gamma$ then, for each $o \in \mathbb{R}^d$*

$$(1 - O(\varepsilon))\text{cost}(Z, o) - O(\varepsilon)\Gamma \leq |Z|\text{cost}(q, o) \leq (1 + O(\varepsilon))\text{cost}(Z, o) + O(\varepsilon)\Gamma$$

*Proof.* To obtain the first inequality, we apply Lemma 19 to bound $\text{cost}(z, o) \leq (1 + \varepsilon)\text{cost}(z, o) + (1 + 1/\varepsilon)\text{cost}(z, q)$ for each $z \in Z$. To obtain the second inequality, we bound $\text{cost}(q, o) \leq (1 + \varepsilon)\text{cost}(z, o) + (1 + 1/\varepsilon)\text{cost}(z, q)$ for each $z \in Z$. $\qquad\square$

**Lemma 21.** *Let $X = \{x_1 \ldots x_\ell\}$ be a weighted set of points in $\mathbb{R}^d$ such that $x_i$ has weight $w_i$. Let $\mu$ be the weighted average of $X$. Let $\hat{\mu} = \hat{\mu}(\alpha_1 \ldots \alpha_\ell)$ be the weighted average of $X$ where $x_i$ has weight $\alpha_i$. If $w_i \leq \alpha_i \leq w_i/(1 - \varepsilon)$ for each $i = 1 \ldots \ell$, then if we interpret $\text{cost}(X, C)$ as $\sum_{x_i \in X} w_i \cdot \text{cost}(x_i, C)$ we have $\text{cost}(X, \hat{\mu}) \leq (1 + O(\varepsilon))\text{cost}(X, \mu)$.*

*Proof.* We note that $\mu$ minimizes the expression $\text{cost}(X, \mu)$. Moreover, $\text{cost}(X, z) \leq \sum_{i=1}^{\ell} \alpha_i \cdot \text{cost}(x_i, z) \leq \text{cost}(X, z)/(1 - \varepsilon)$. Since $\hat{\mu}$ minimizes the expression $\sum_{i=1}^{\ell} \alpha_i \cdot \text{cost}(x_i, z)$ it must be $\text{cost}(X, \hat{\mu}) \leq \text{cost}(X, \mu)/(1 - \varepsilon)$. $\qquad\square$

Adopting the same proof strategy, we obtain the following.

**Observation 22.** *Thanks to Lemma 20, we can assume that the points in $Z$ are concentrated in $q$ for the purpose of computing a $(1 + O(\varepsilon))$-approximation to the 1-means problem on $Z$, whenever an additive error $\Gamma$ is tolerable. Indeed, moving all points in $Z$ to $q$ introduces a $1 + O(\varepsilon)$ multiplicative error on $\text{cost}(Z, \cdot)$ and a $O(\varepsilon)\Gamma$ additive error.*

The next lemma shows that a point $z$ that is far from a center $o$ experiences a small variation of $\text{cost}(z, o)$ when the position of $o$ is slightly perturbed.

**Lemma 23.** *Given $o, z \in \mathbb{R}^d$ such that $\|o - z\| \geq r/\varepsilon$ we have that for every $o' \in B(o, r)$, $\text{cost}(z, o') = (1 \pm O(\varepsilon))\text{cost}(z, o)$.*

*Proof.* It is enough to prove it for all $o'$ that lie on the line $L$ passing through $o$ and $z$, any other point in $o'' \in B(o, r)$ admits a point $o' \in B(o, r) \cap L$ with $\|o' - z\| = \|o'' - z\|$. It is enough to compute the derivative of $\text{cost}(z, \cdot)$ with respect to the direction of $L$ and see that $\frac{\partial \text{cost}(z, \cdot)}{\partial L}|_{B(o,r)} = (1 \pm O(\varepsilon))r/\varepsilon$. Thus, $\text{cost}(z, o') = \text{cost}(z, o) \pm (1 \pm O(\varepsilon))r^2/\varepsilon = (1 \pm O(\varepsilon))\text{cost}(z, o)$. $\qquad\square$

*Proof of Lemma 15.* Here we prove that for each $o_1 \ldots o_s$ there exist coefficients $\alpha_0^{(i)} \ldots \alpha_s^{(i)} \in \left\{1, (1 - \varepsilon) \ldots (1 - \varepsilon)^{\lceil \log_{1-\varepsilon}(\varepsilon^7) \rceil}\right\} \cup \{0\}$ such that the convex combination $\hat{o}_i = \hat{o}_i(\alpha_0^{(i)} \ldots \alpha_s^{(i)})$ is a perfect approximation of $o_i$, with probability $k^{-O(\varepsilon^{-2})} \cdot 2^{-poly(\varepsilon^{-1})}$. Wlog, we show this

for $o_1$ only. Concretely, we want to show that, with probability $k^{-O(\varepsilon^{-1})} \cdot 2^{-poly(\varepsilon^{-1})}$, there exist coefficients $\alpha_0 \ldots \alpha_s$ such that $\hat{o}_1 = \hat{o}_1(\alpha_0 \ldots \alpha_s)$ satisfies $\texttt{cost}(O_1^*, (\mathcal{C} \cup \{\hat{o}_1\}) \setminus Out) \leq (1 + O(\varepsilon))\text{OPT}_1 + O(\varepsilon)\text{OPT}/k$. Taking the joint probability of these events for each $i = 1 \ldots s$ we obtain the success probability $k^{-O(\varepsilon^{-2})} \cdot 2^{-poly(\varepsilon^{-1})}$. Note that we are supposed to prove that $\texttt{cost}(O_1^*, (\mathcal{C} \cup \{\hat{o}_1\}) \setminus Out) \leq (1 + \varepsilon)\text{OPT}_1 + \varepsilon\text{OPT}/k$, however we prove a weaker version where $\varepsilon$ is replaced by $O(\varepsilon)$, which is in fact equivalent up to rescaling $\varepsilon$.

Similarly to $\mathcal{C}[\cdot]$ and $\mathcal{O}^*[\cdot]$ define $\mathcal{T}[p]$ as the closest center to $p$ in $\mathcal{T}$. Denote with $C_1$, $F_1$ and $N_1$ the intersections of $O_1^*$ with $C$, $F$ and $N$ respectively. In what follows we define the values of $\alpha_0 \ldots \alpha_s$ that define $\hat{o}_1 = \hat{o}_1(\alpha_0 \ldots \alpha_s)$ and show an assignment of points in $O_1^*$ to centers in $(\mathcal{C} \cup \{\hat{o}_1\}) \setminus Out$ with cost $(1 + O(\varepsilon))\text{OPT}_1 + O(\varepsilon)\text{OPT}/k$. Recall that we assume that $q_i \in \texttt{core}(o_i)$ for each $i = 1 \ldots s$.

In what follows, we assign values to the coefficients $(\alpha_i)_i$. It is understood that if the final value we choose for $\alpha_i$ is $v$ then we rather set $\alpha_i$ to the smallest power of $(1 - \varepsilon)$ which is larger than $v$, if $v > \varepsilon^7$. Else, set $\alpha_i$ to 0. We will see in the end that this restrictions on the values of $\alpha_i$ do not impact our approximation.

In what follows, we will assign the points in $O_1^*$ to $\mathcal{C} \setminus Out$, if this can be done inexpensively. If it cannot, then we will assign points to $\hat{o}_1$. In order to compute a good value for $\hat{o}_1$ we need an estimate of the average of points assigned to $\hat{o}_1$. For points in $N_1$, computing this average is doable (leveraging Lemma 13) while for points in $O_1^* \setminus N_1$ we show that either their contribution is negligible or we can collapse them so as to coincide with some $q_i \in \mathcal{Q}$ without affecting our approximation. The coefficients $(\alpha_i)_{i \geq 1}$ represent the fraction of points in $O_1^*$ which is collapsed to $q_i$. $\alpha_0$ represents the fraction of points in $O_1^*$ which average we estimate as $\mu(S)$. Thus, Equation (3) defines $\hat{o}_i$ as the weighted average of points $q_i$, where the weights are the (approximate) fractions of points collapsed onto $q_i$, together with the the average $\mu(S)$ and its associated weight $\alpha_0$.

**Points in $C_1$.** All points $p \in C_1$ such that $\mathcal{T}[p] \notin \mathcal{Q}$ can be assigned to $\mathcal{T}[p] \in \mathcal{C} \setminus Out$ incurring a total cost of at most $\varepsilon^6\text{OPT}_1$, by the definition of $C_1$. Given a point $p \in C_1$ with $\mathcal{T}[p] \in \mathcal{Q}$ we might have $\mathcal{T}[p] \notin \mathcal{C} \setminus Out$ and thus we cannot assign $p$ to $\mathcal{T}[p]$. Denote with $W$ the set of points $p$ with $\mathcal{T}[p] \in \mathcal{Q}$. Our goal is now to approximate $\mu(W)$. In order to do that, we will move each $p \in W$ to coincide with $q_i = \mathcal{T}[p]$. We can partition $W$ into $W_1 \ldots W_s$ so that for each $z \in W_i$ $\mathcal{T}[z] = q_i$. If $p \in Z_i$ then we have $||p - q_i||^2 \leq \varepsilon^3\rho_1^2$. Hence, thanks to Observation 22, we can consider points in $W_i$ as if they were concentrated in $q_i$ while losing at most an additive factor $O(\varepsilon)\text{OPT}_1$ and a multiplicative factor $(1 + \varepsilon)$ on their cost. For $i = 1 \ldots s$, set $\alpha_i \leftarrow |W_i|/|O_1^*|$. In this way, $\sum_{i=1}^s \alpha_i \cdot q_i / \sum_{i=1}^s \alpha_i$ is an approximates solution to 1-mean on $W$ up to a multiplicative factor $(1 + \varepsilon)$ and an additive factor $O(\varepsilon)\text{OPT}_1$.

**Points in $N_1$.** Consider the two cases: $(i)$ $\texttt{cost}(N_1, \mathcal{T}) > \varepsilon^2\text{OPT}/k$; $(ii)$ $\texttt{cost}(N_1, \mathcal{T}) \leq \varepsilon^2\text{OPT}/k$.

Case $(i)$. We show that in this case $\mu(S)$ is a $(1 + \varepsilon)$-approximation for 1-mean on $N_1$, with probability $k^{-O(\varepsilon^{-1})} \cdot 2^{-poly(\varepsilon^{-1})}$. First, notice that if we condition on $S \subseteq N_1$ then Lemma 13 gives that $\mu(S)$ is a $(1 + \varepsilon)$-approximation for 1-mean on $N_1$ with constant probability. Thus, we are left to prove that $S \subseteq N_1$ with probability $k^{-O(\varepsilon^{-1})} \cdot 2^{-poly(\varepsilon^{-1})}$. We have that the $P_{p \sim \texttt{cost}(p, \mathcal{T})}[p \in N_1 \mid p \in N] \geq \varepsilon^2/k$, however the costs w.r.t. $\mathcal{T}$ of points in $N$ varies of at most a factor $poly(\varepsilon^{-1})$, thus $P_{p \sim Unif}[p \in N_1 \mid p \in N] \geq poly(\varepsilon)/k$. The probability of $S \subseteq N_1$ is thus $(poly(\varepsilon)/k)^{|S|} = k^{-O(\varepsilon^{-1})} \cdot 2^{-poly(\varepsilon^{-1})}$. In this case, we set $\alpha_0 \leftarrow |N_1|/|O_1^*|$ because $\mu(S)$ approximates the mean of the entire set $N_1$.

Case $(ii)$. Here we give up on estimating the mean of $N_1$ and set $\alpha_0 \leftarrow 0$. The point $x \in N_1$ such that $\mathcal{T}[x] \notin \mathcal{Q}$ can be assigned to $\mathcal{T}[x]$ incurring a combined cost of $\varepsilon^2\text{OPT}/k$. We partition the remaining points in $N_1$ into $Z_1 \cup \ldots Z_s$ where each point $x$ is placed in $Z_i$ if $\mathcal{T}[x] = q_i$. Now, we collapse the points in $Z_i$ so as to coincide with $q_i$ and show that this does not worsen our approximation factor. In terms of coefficients $(\alpha_i)_i$, this translates into the updates $\alpha_i \leftarrow \alpha_i + |Z_i|/|O_1^*|$ for each $i = 1 \ldots s$.

Indeed, using Observation 22 we can move all points in $Z_i$ to $q_i$ incurring an additive combined cost of $\varepsilon\text{OPT}/k$ and a multiplicative cost of $1 + O(\varepsilon)$.

**Points in $F_1$.** Points in $F_1$ are very far from $q_1$ and thus far from $o_1$, hence even if their contribution to $\texttt{cost}(O_1^*, o_1)$ might be large, we have $\texttt{cost}(F_1, o_1) = (1 \pm O(\varepsilon))\texttt{cost}(F_1, o')$ for all $o'$ in a ball of radius $\rho_1/\varepsilon$ centered in $o_1$, thanks to Lemma 23.

Let $H$ be the set of points that have not been assigned to centers in $\mathcal{C} \setminus Out$. In particular, $H = W \cup N_1$ if points in $N_1$ satisfy case $(i)$ and $H = W \cup Z_1 \dots Z_s$ if points in $N_1$ satisfy case $(ii)$. We consider two cases.

If $||\mu(H) - q_1|| \leq \rho/\varepsilon$, then $||\mu(H) - o_1|| \leq \rho(1 + \varepsilon + 1/\varepsilon)$ because $q_1 \in \texttt{core}(o_1)$. Since for each $f \in F_1$ we have $||f - o_1|| \geq ||f - q_1|| - (1 + \varepsilon)\rho \geq \Omega(\rho/\varepsilon^3)$ then $\texttt{cost}(f, o') = (1 \pm O(\varepsilon))\texttt{cost}(f, o_1)$ for each $o'$ in a ball of radius $O(\rho/\varepsilon)$ centered in $o_1$, and so in particular for $o' = \mu(H)$. Thus in this case we can simply disregard all points in $F_1$ and computing $\hat{o}_1$ according to the $(\alpha_i)_i$ defined above yields a perfect approximation of $o_i$.

Else, if $||\mu(H) - q_1|| > \rho/\varepsilon$, a similar argument applies to show that $\texttt{cost}(H, o') = (1 \pm \varepsilon)\texttt{cost}(H, o)$ for each $o'$ in ball of radius $O(\rho)$ centered in $o_1$. Indeed, we can rewrite $\texttt{cost}(H, o')$ as $|H| \cdot \texttt{cost}(\mu(H), o') + \texttt{cost}(\mu(H), H)$. If $||\mu(H) - q_1|| < \rho/\varepsilon$ the first term varies of at most a factor $(1 + \varepsilon)$ and the second term is constant. Thus in this case $\hat{o}_1 = q_1$ is a perfect approximation of $o_1$ and we simply set $\alpha_1 = 1$ and $\alpha_j = 0$ for $j \neq 1$. In other words, here $\mu(N_1 \cup H)$ is too far from $q_1$ (and thus $o_1$) to significantlyt influence the position of $\hat{o}_1$ and the same holds for any point in $F_1$. This works, of course, because we assumed $q_1 \in \texttt{core}(o_1)$. $\qquad\square$

**Discussing the limitations on the coefficients values.** The proof above would work smoothly if we were allowed to set $\alpha_i$ to exactly the values discussed above, representing the fractions of points from $O_i^*$ captured by different $q_i$s. However, to make the algorithm efficient we limit ourselves to values in $\left\{1, (1 - \varepsilon) \dots (1 - \varepsilon)^{\lceil \log_{1-\varepsilon}(\varepsilon^7) \rceil}\right\} \cup \{0\}$. Lemma 21 shows that as long as the values of $(\alpha_i)_i$ estimate the frequencies described above up to a factor $1 \pm O(\varepsilon)$ then the approximation error is within a multiplicative factor $1 \pm O(\varepsilon)$.

We are left to take care of the case in which $\alpha_i$ is set to a value $< \varepsilon^7$. We set $\alpha_i$ when dealing with points in $C_1 \cup N_1$ and for each $x \in C_1 \cup N_1$ we have, for each $o' \in B(q_1, (1 + \varepsilon)\rho)$, $\texttt{cost}(x, o') \leq 2\texttt{cost}(q_1, o') + 2\texttt{cost}(x, q_1) = O(\rho_1\varepsilon^{-6})$. Thus, if we simply set $\alpha_i \leftarrow 0$ whenever we have $\alpha_i < \varepsilon^7$ then the combined cost of points in $O_1^*$ with respect to $o'$ varies by $\varepsilon^7 |O_1^*| \cdot \rho_1 \varepsilon^{-6} = O(\varepsilon)\text{OPT}_1$. Effectively, ignoring these points does not significantly impact the cost. hence solving 1-mean ignoring these points finds a $(1 + O(\varepsilon))$-approximate solution to the original problem.

## Additional Experimental Evaluation

In this section we report additional experiments which presentation did not fit in the main body. In particular, we run experiments on the dataset KDD-PHY and for $k = 10, 50$.

In Figure 4 we compare MSLS-G with MSLS. To perform our experiment, we initialize $k = 25$ centers using KM++ and then run $50$ iterations of local search for both algorithms, for $p \in \{2, 3\}$ swaps. We repeat each experiment 5 times. For ease of comparison, we repeat the plot for the KDD-BIO and RNA datasets that we present in the main body of the paper. Due to the higher running of the MSLS we perform this experiments on 1% uniform sample of each of our datasets. We find out that the performance of the two algorithms is comparable on all our instances, while they both perform roughly 15%-27% better than $k$-means++ at convergence.

In Figure 5 we run KM++ followed by $50$ iterations of MSLS-G with $p = 1, 4, 7, 10$ and $k = 10, 25, 50$ (expcluding the degenerate case $p = k = 10$) and plot the relative cost w.r.t. KM++ at each iteration. The results for $k = 25$ on KDD-BIO and RNA can be found in Figure 2. We repeat each experiment 5 times. Our experiment shows that, after 50 iterations MSLS-G for $p = 4, 7, 10$ achieves improvements of roughly $5 - 10\%$ compared to MSLS-G-$p = 1$ and of the order of $20\% - 40\%$ compared to KM++. These improvements are more prominent for $k = 25, 50$. We also report the time per iteration that each algorithm takes. For comparison, we report the running time of a single iteration of Lloyd's next to the dataset's name. Notice that the experiment on RNA for $k = 50$ is performed on a 10% uniform sample of the original dataset, due to the high running time.

In Figure 6, we use KM++ and MSLS-G as a seeding algorithm for Lloyd's and measure how much of the performance improvement measured is retained after running Lloyd's. First, we initialize

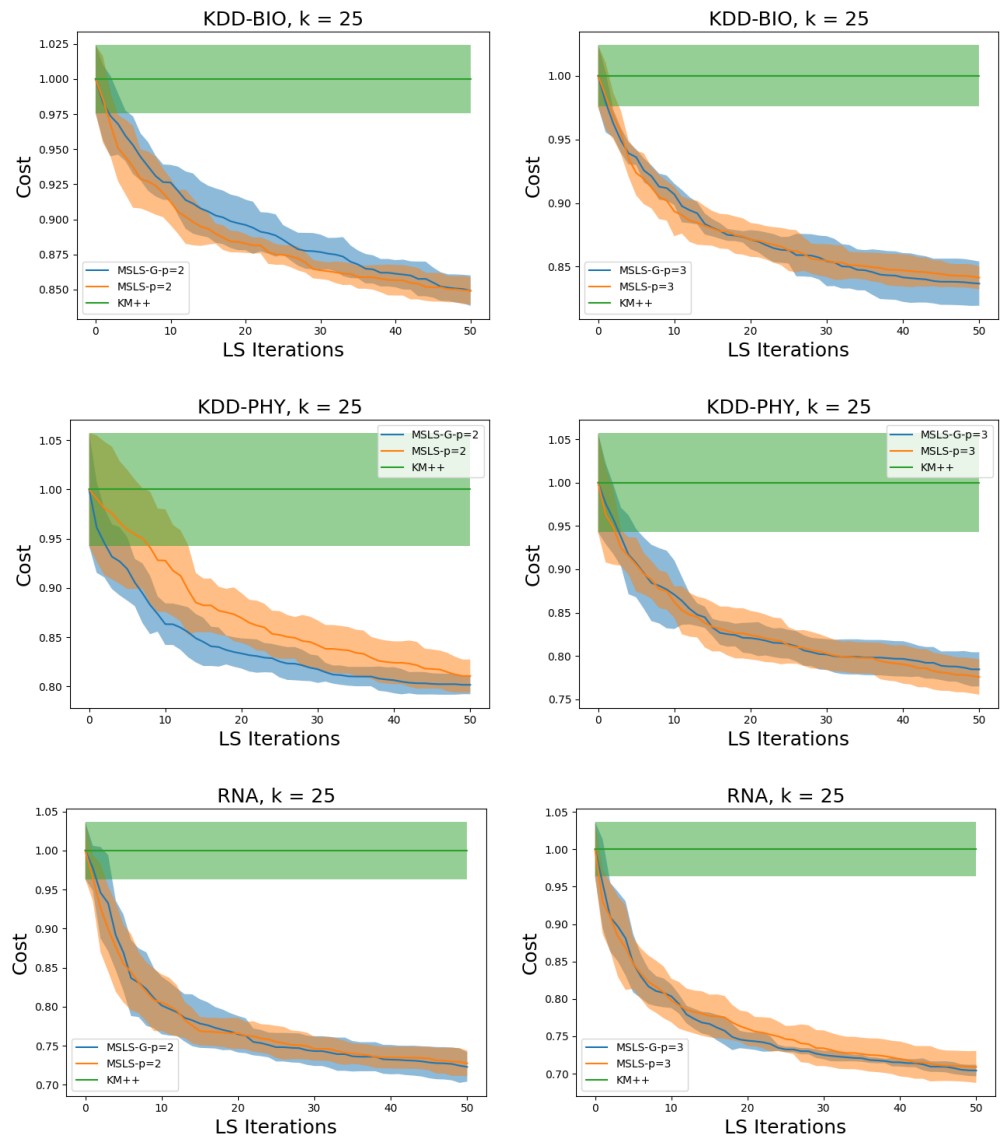

Figure 4: Comparison between MSLS and MSLS-G, for $p = 2$ (left column) and $p = 3$ (right column), for $k = 25$, on the datasets KDD-BIO (first row), KDD-PHY (second row) and RNA (third row). The $y$ axis shows the mean solution cost, over the 5 repetitions of the experiment, divided by the means solution cost of KM++.

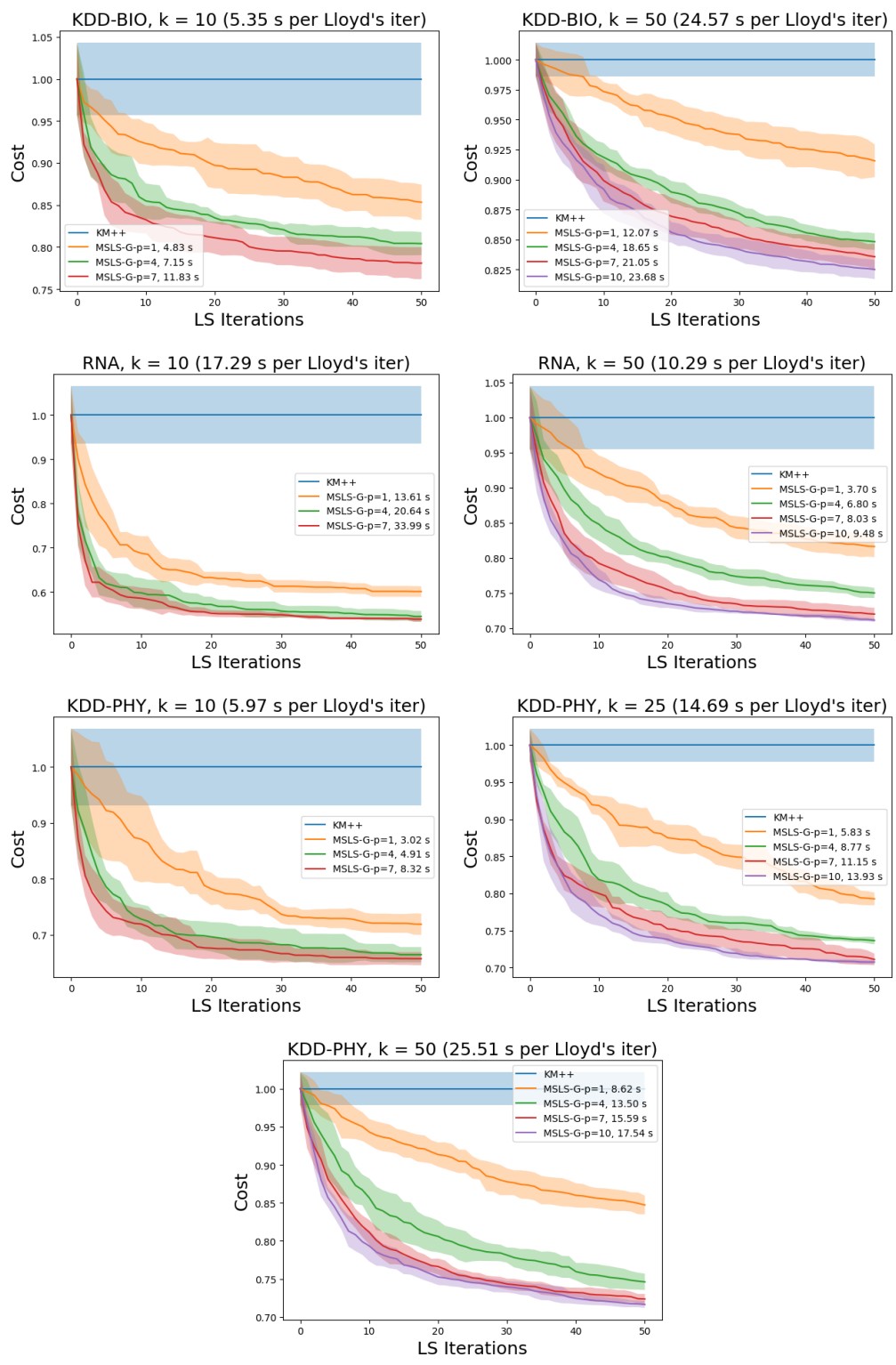

Figure 5: We compare the cost of MSLS-G, for $p \in \{1, 4, 7, 10\}$, divided by the mean cost of KM++ at each LS step, for $k \in \{10, 25, 50\}$, excluding the degenerate case $p = k = 10$. The legend reports also the running time of MSLS-G per LS step (in seconds). The experiments were run on all datasets: KDD-BIO, RNA and KDD-PHY, excluding the case of $k = 25$ for KDD-BIO and RNA which are reported in the main body of the paper.

our centers using KM++ and the run $15$ iterations of MSLS-G for $p = 1, 4, 7$. We measure the cost achieved by running $10$ iterations of Lloyd's starting from the solutions found by MSLS-G as well as KM++. We run experiments for $k = 10, 25, 50$ and we repeat each experiment $5$ times. We observe that for $k = 25, 50$ MSLS-G for $p > 1$ performs at least as good as SSLS from Lattanzi and Sohler [2019] and in some cases maintains non-trivial improvements. These improvements are not noticeable for $k = 10$; however, given how Lloyd's behave for $k = 10$ we conjecture that $k = 10$ might be an "unnatural" number of clusters for our datasets.

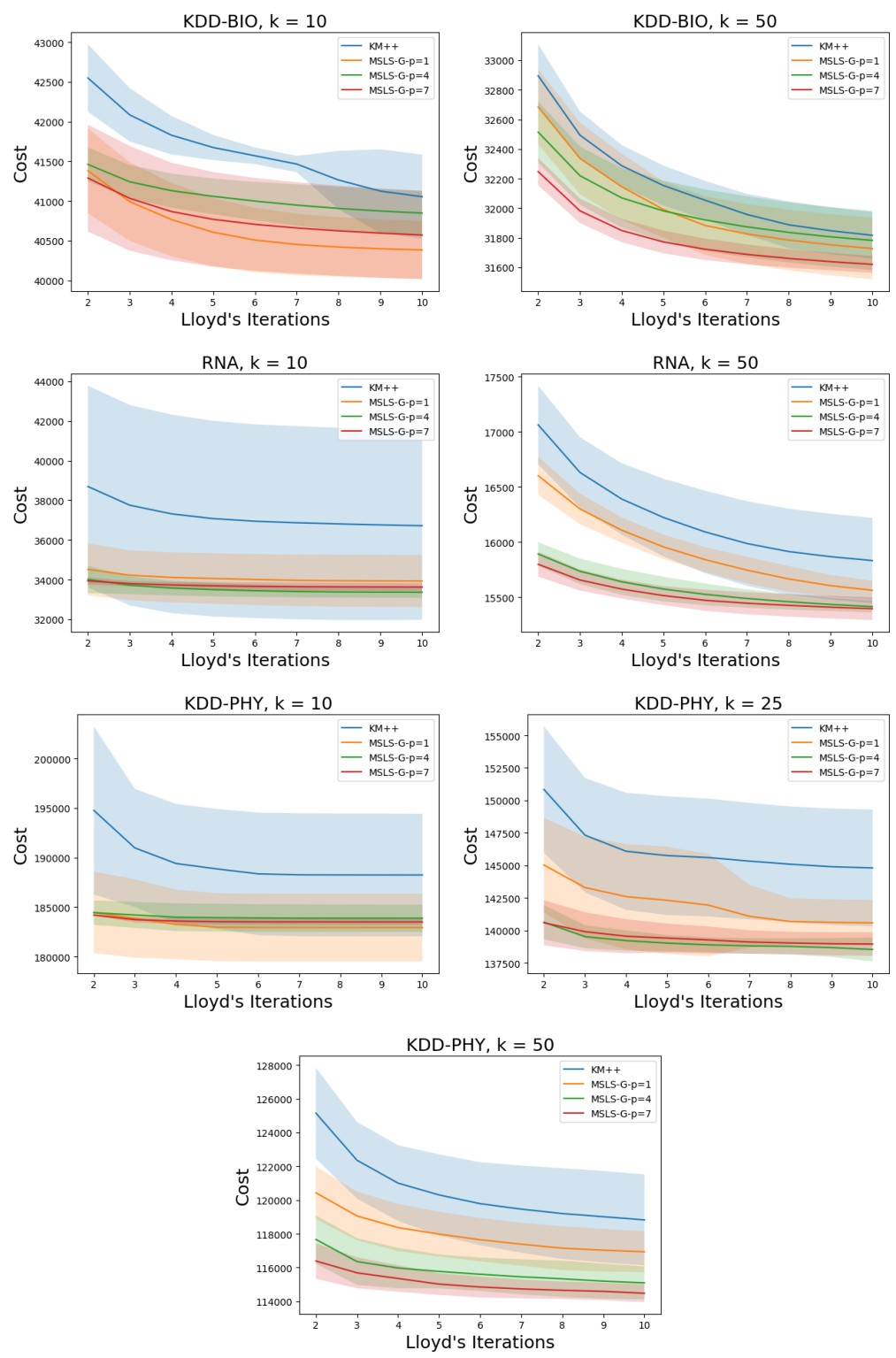

Figure 6: We compare the cost after each of the 10 iterations of Lloyd with seeding from MSLS-G, for $p \in \{1, 4, 7, 10\}$ and 15 local search steps and KM++, for $k \in \{10, 25, 50\}$. We excluded the degenerate case $p = k = 10$, and the experiments reported in the main body of the paper.

