# Multi-Swap $k$-Means++

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

set of weighted ideal swaps $\mathcal{S}$, if $o_i$ belongs to a $s$-swap $(In, Out) \in \mathcal{S}$ for $s > 1$ then it is good if and only if $(In, Out)$ is good; else $o_i$ might belong to up to $p-1$ single swaps and then any of them being good would suffice. Denote with $G$ the union of cores of good optimal centers in $\widetilde{\mathcal{O}}$.

**Lemma 10.** *If an ideal swap $(In, Out)$ is bad, then we have*

$$\mathtt{cost}(O_{In}^*, \mathcal{C}) \leq (2+\delta)\mathtt{cost}(O_{In}^*, \mathcal{O}^*) + \mathtt{Reassign}(In, Out) + \delta\mathrm{ALG}/k. \qquad (2)$$

*Proof.* Let $In = \{o_1 \dots o_s\}$, $\mathcal{Q} = \{q_1 \dots q_s\}$ such that $q_1 \in \mathtt{core}(o_1) \dots q_s \in \mathtt{core}(o_s)$. Then, by Lemma 1 $\mathtt{cost}(O_{In}^*, \mathcal{Q}) \leq (2+\delta)\mathtt{cost}(O_{In}^*, \mathcal{O}^*)$. Moreover, $\mathtt{Reassign}(In, Out) = \mathtt{cost}(P \setminus O_{In}^*, \mathcal{C} \setminus Out) - \mathtt{cost}(P \setminus O_{In}^*, \mathcal{C})$ because points in $P \setminus C_{Out}$ are not affected by the swap. Therefore, $\mathtt{cost}(P, (\mathcal{C} \cup \mathcal{Q}) \setminus Out) \leq (2+\delta)\mathtt{cost}(O_{In}^*, \mathcal{O}^*) + \mathtt{Reassign}(In, Out) + \mathtt{cost}(P \setminus O_{In}^*, \mathcal{C})$. Suppose by contradiction that Equation (4) does not hold, then

$$\mathtt{cost}(P, \mathcal{C}) - \mathtt{cost}(P, (\mathcal{C} \cup \mathcal{Q}) \setminus Out) =$$
$$\mathtt{cost}(P \setminus O_{In}^*, \mathcal{C}) + \mathtt{cost}(O_{In}^*, \mathcal{C}) - \mathtt{cost}(P, (\mathcal{C} \cup \mathcal{Q}) \setminus Out) \geq \delta\mathrm{ALG}/k.$$

Hence, $(\mathcal{Q}, Out)$ is strongly improving and this holds for any choice of $\mathcal{Q}$, contradiction. $\qquad \square$

**Lemma 11.** *If $\mathrm{ALG}/\mathrm{OPT} > \eta^2 + \delta$ then $\mathtt{cost}(G, \mathcal{C}) = \Omega_\delta(\mathtt{cost}(P, \mathcal{C}))$. Thus, if we $D^2$-sample $q$ we have $P[q \in G] = \Omega_\delta(1)$.*

*Proof.* First, we observe that the combined current cost of all optimal clusters in $\mathcal{O}^* \setminus \widetilde{\mathcal{O}}$ is at most $k \cdot \delta\mathrm{ALG}/k = \delta\mathrm{ALG}$. Now, we prove that the combined current cost of all $O_i^*$ such that $o_i$ is bad is $\leq (1 - 2\delta)\mathrm{ALG}$. Suppose, by contradiction, that it is not the case, then we have:

$$(1 - 2\delta)\mathrm{ALG} < \sum_{\text{Bad } o_i \in \widetilde{\mathcal{O}}} \mathtt{cost}(O_i^*, \mathcal{C}) \leq \sum_{\text{Bad } (In, Out) \in \mathcal{S}} w(In, Out) \cdot \mathtt{cost}(O_{In}^*, \mathcal{C}) \leq$$

$$\sum_{\text{Bad } (In, Out)} w(In, Out) \cdot ((2+\delta)\mathtt{cost}(O_{In}^*, \mathcal{O}^*) + \mathtt{Reassign}(In, Out) + \delta\mathrm{ALG}/k) \leq$$

$$(2+\delta)\mathrm{OPT} + (2 + 2/p)\mathrm{OPT} + (2 + 2/p)\sqrt{\mathrm{ALG}}\sqrt{\mathrm{OPT}} + \delta\mathrm{

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

 \texttt{cost}(x_i, C)$ *we have $\texttt{cost}(X, \hat{\mu}) \leq (1 + O(\varepsilon))\texttt{cost}(X, \mu)$.*

638    *Proof.* We note that $\mu$ minimizes the expression $\texttt{cost}(X, \mu)$. Moreover, $\texttt{cost}(X, z) \leq \sum_{i=1}^{\ell} \alpha_i \cdot$
639    $\texttt{cost}(x_i, z) \leq \texttt{cost}(X, z)/(1 - \varepsilon)$. Since $\hat{\mu}$ minimizes the expression $\sum_{i=1}^{\ell} \alpha_i \cdot \texttt{cost}(x_i, z)$ it
640    must be $\texttt{cost}(X, \hat{\mu}) \leq \texttt{cost}(X, \mu)/(1 - \varepsilon)$.    $\