# OpenReview forum: "Multi-Swap k-Means++"
_NeurIPS.cc/2023/Conference — NeurIPS 2023 poster_

### Official Review · Reviewer_coFq · 2023-06-25

**Soundness:** 3 good
**Presentation:** 4 excellent
**Contribution:** 4 excellent
**Rating:** 8
**Confidence:** 3

**Summary:**

This paper studies local-search algorithms for k-means clustering. The goal here is to obtain a local-search algorithm which (1) give a close to 9-approximation ratio (which is best possible for local search algorithms) and (2) is practical. In the past literature, there has been many local search algorithms developed for the k-means problem, but all of them either have a relatively big approximation ratio or have a prohibitive running time in practice due to the complexity of the local moves at each step.

In this paper, the authors give a simple local search algorithm which is practical and guarantees a 10.48-approximation. They also give a more complex local search which matches the approximation lower bound of 9, but with a significantly faster running time than in Kanungo et al. [2004]. However, this algorithm is significantly more complex and is unlikely to be used in practice. They also give experiments that show that their first algorithm and a slight variant of it perform well in practice.


**Strengths:**

In my opinion, the strengths are as follows.
1) The paper is clearly written.
2) The first algorithm is a nice result since it combines practicality with an almost best possible theoretical guarantee. As far as I know (not being an expert), this is the first time the D^2-sampling is used and analyzed in combination with multi-swap local search. Overall, the techniques in the paper are nice.

**Weaknesses:**

I do not see an obvious weakness in this work, the results are tight or close. Overall, I think this is a nice paper that fits well in the context of NeurIPS.

**Questions:**

It is unclear to me if the approximation ratio of 10.48 is tight for the algorithm. As well as the 26.64 for the single swap version.

In experiment section, I do not see any discussion on the running time of the compared algorithms? Maybe it would be nice just to mention it.

**Limitations:**

Yes.

---

> ### Author Rebuttal · Authors · 2023-08-09
>
> We thank the reviewer for their time and effort put into reviewing the paper.
>
> Answer to the questions:
>
> We did not manage to prove the tightness of our analysis, neither for $p = 1$ (26.64..), nor for $p$ approaching infinity (10.48..). This is an interesting open question. However, it seems to us that this should be the case. Indeed, if we could consider the optimal centers as candidate centers to swap in, then the term $2 + \delta$ in Eq. (2) would turn into a $1 + \delta$. If that term is $1 + \delta$, then the analysis gives a $9 + \epsilon$ approximation ratio for p sufficiently large. This suggests that, allowing only data points is actually the only factor holding our algorithm back from achieving optimal approximation ratio. Moreover, the factor $2 + \delta$ in Eq (2) is best possible: think of a cluster where all data points lie on a sphere centered in their optimal center. Thus, we should expect the analysis to be tight, at least for $p$ approaching infinity. All this is a bit speculative, and we have no formal proof that our analysis is actually tight.
>
>
> We kindly note that the running times of the algorithms (per iteration) are reported in the legends of Figure 2 following the algorithm names. We realized this was confusing, and we will report the running times more prominently in the next version of our paper.

---

> > ### Comment · Reviewer_coFq · 2023-08-14
> >
> > I would like to thank the authors for their reponse and clarifications. After looking at other reviews and rebuttals, my initial assessment remains. I think this is a nice paper.

---

### Official Review · Reviewer_2sgw · 2023-07-02

**Soundness:** 2 fair
**Presentation:** 2 fair
**Contribution:** 2 fair
**Rating:** 3
**Confidence:** 4

**Summary:**

This paper studies the standard $k$-means problem. Given a $k$-means instance $(P,k)$, the goal is to find a set $C$ of centers with size at most $k$ such that the sum of the squared  distances from $P$ to $C$ is minimized. For the $k$-means problem, Lattanzi and Sohler (ICML 2019) proposed an elegant combination of the local search and the $k$-means++ seeding methods. Instead of enumerating all the swap pairs constructed between the data points and the current centers, they showed that $k$-means++ sampling can be used for sampling a data point to serve as the candidate center for swapping in such that a $509$-approximation can be achieved in expectation in time $O(ndk^2loglogk)$. This paper extends the single-swap strategy to multi-swap strategy. In a single local search step, this paper shows that using $k$-means++ to sample $t$ data points induces a good $t$-swap for the local search process such that the clustering cost can be reduced significantly by at least a $\Omega(1-1/k)$ fraction with certain probability. After $O(k^{t-1})$ local search steps, an improved approximation can be achieved related to the swap size $t$, where the total running time can be bounded by $\tilde{O}(ndk^{2t})$. When $t$ is large enough, the approximation could be smaller than 10.48. By combining the techniques from Cohen Addad et al. (NeurIPS 2021), the papers shows that by using more local search steps (with exponential dependence on $\epsilon$), the approximation guarantee can nearly match the lower bound of the $9+\epsilon$ of the standard multi-swap local search method.

**Strengths:**

The strengths of this paper can be summarized as follows:
1. This paper proposes a fast multi-swap local search method for solving the standard $k$-means problem, which runs in linear time in the data size. The approximation ratio significantly improves the previous one with linear running time in the data size (i.e., 509-approximation)
2. This paper gives another multi-swap local search method which better approximates the optimal clustering centers during the local search swaps using a subroutine called APX-CENTERS, where an improved approximation guarantee, i.e., $9+\epsilon$, can be obtained with running time exponentially dependent on $poly(\epsilon^{-1})$. The approximation ratio matches the approximation lower bound of local search methods for the $k$-means problem and significantly improves the running time of previous work based on direct enumeration of swap pairs constructed between the whole dataset and the current centers opened.
3. In experiments, this paper proposes a heuristic method which avoids the exhaustive searching process for determining which subset of the sampled set of data points should be swapped in. The experiments show that the multi-swap local search method achieves better performance on clustering quality with fixed local search iterations compared with single-swap local search methods.

**Weaknesses:**

1. Although the authors show that sampling-based multi-swap local search can improve the approximation to a very small constant and even matches the lower bound of local search methods, the core idea behind is to use the successive $k$-means++ sampling strategy to construct candidate set of centers for swapping in, which is an extension of the sampling-based local search method (denoted as LS++ method )proposed by Lattanzi and Sohler. The claimed improvement in ratio seems to be minor as LS++ already guarantees constant approximation (in [1] the authors claimed that they did not attempt to optimize the constants and this paper verifies that LS++ can yield approximation ratio smaller than 26.4). The key idea behind the analysis for multi-swap local search method is as follows: (1) the authors further divide the optimal clusters into different groups according to their current clustering cost (i.e., $cost(P_h^*,C) \ge \delta ALG/k$ or $cost(P_h^*,C) < \delta ALG/k$) (2) the authors give weights for swap pairs such that each current center can be used for at most $(1+1/t)$ times when performing a summation of all bad swap pairs. The analysis for probability lower bound that induces a good swap is similar to that of LS++ method.

2. Some theoretical details is unclear to me. During the construction of ideal weighted multi-swaps, this paper partitions the optimal clustering centers and current clustering centers based on $\tilde{O}=${
$o_i \ | \ cost(O_i^*,C) \ge \delta ALG/k$} and $\tilde{C}=$ $C\backslash$ {$C[o_i] \ | \ o_i \in O^* \backslash \tilde{O}$}. Then, in the construction of ideal multi-swaps, each swap is consist of $In$ and $Out$ such that $In \subseteq \tilde{O}$ and $Out \subseteq \tilde{C}$. If my understanding is not wrong, some of the optimal centers in $\tilde{O}$ is not used for construction of ideal multi-swaps. If not all optimal centers is used at least once for constructing ideal multi-swaps, then the bound for summation of clustering cost for all "bad'" optimal clusters may not hold in Lemma 11. The ideal multi-swaps construction considers the two caes: (1) for each center $c_i \in \tilde{C}$ that is neither busy nor lonely, form $In$ with the optimal centers captured by $c_i$ (multi-swap) and borrow $|In|-1$ lonely centers from $L$ (2) for each center in $\tilde{C}$ that is busy, form single swap using each of optimal centers captured by $c_i$ and the lonely centers in $L$. However, there may exist some case that for some optimal clusering center $o_i^* \in \tilde{O}$, the current center $c_j$ that captures $o_i^*$ is not in $\tilde{C}$ and hence $o_i^*$ will no longer be used for constructing ideal multi-swaps. Here is an example where $o_i^*, o_2^*, o_3^*$ is the optimal clustering centers, and $c_1$, $c_2$, $c_3$ is the current centers with swap size $p=2$. In this instances, it holds that $c_1$ captures $o_1^*$ and $o_2^*$, $c_3$ captures $o_3^*$, $cost(O_1^*,C) \ge \delta ALG/k$, $cost(O_2^*,C)<\delta ALG/k$ and $cost(O_3^*,C) \ge \delta ALG/k$. Hence, we have $\tilde{O}=${$o_1^*,o_3^*$} and  $\tilde{C} =${$c_2,c_3$}. During the construction of ideal multi-swaps, since $c_1$ is not in $\tilde{C}$, $o_1^*$ will never be used during the construction.

3. Another main weakness of this paper is the experiments. (1) As a local search algorithm, it’s important to validate the performance of the proposed algorithm within a certain time limitation, which is not presented in the experimental results. This paper mainly compares the experimental performances with fixed local search steps. It is unfair since multi-swap local search method takes more time than single-swap local search methods (i.e., the LS++ method). (2) In experiments, the maximum local search step is set to be 50, which is much smaller than the theoretical bounds (100000kloglogk) for obtaining good performance for LS++ algorithm. For this setting, even single-swap local search method may not converge to a good local optimal solution. The authors should present more evaluation for larger local search steps (i.e., several hundreds or thousands of local search steps) and fixed running time that is large enough for each algorithm to reach the convergence. (3) The size of the tested datasets is rather small (i.e., no larger than 500,000). However, in the recent results for clustering algorithms, instances of size over 1 million [1] or even 100 million [2] have been considered. Since multi-swap local search is much slower than the single-swap local search method (even with heuristic acceleration by avoiding the enumeration of all subsets of data points for swapping in), it is unclear that whether the proposed algorithm can scale well on large-scale datasets compared with single-swap local search method.

[1] Ren J, Hua K, Cao Y. Global Optimal K-Medoids Clustering of One Million Samples[J]. Advances in Neural Information Processing Systems, 2022, 35: 982-994.
[2] Matsui Y, Ogaki K, Yamasaki T, et al. Pqk-means: Billion-scale clustering for product-quantized codes[C]//Proceedings of the 25th ACM international conference on Multimedia. 2017: 1725-1733.

**Questions:**

1. During the construction of ideal multi-swaps, can the construction process described in the paper guarantee that all the optimal centers in $\tilde{O}$ is used at least once for constructing ideal multi-swaps (see the example of weakness 2). If some of the optimal clustering centers   are not used for constructing ideal multi-swaps, will the bound for summation of clustering cost for all "bad'" optimal clusters in Lemma 11 still hold?

2. In Lemma 2, this paper shows that if the current clustering cost for some optimal cluster $O_i^*$ is larger than $(2+3\delta)$ times its optimal clustering cost, the clustering cost of data points close to $o_i^*$ should take a large fraction (related to $\delta$) of the whole clustering cost induced by $O_i^*$. Hence, in each local search step, the success probability for performing a good swap should be related to parameter $\delta$, for example $\Omega(1/\delta)$. However, the running time of the proposed multi-swap local search algorithm is independent of $\delta$. I think there should be at least an $\frac{1}{\delta}$ term in the running time, or does it mean that $\delta$ is a constant given as the input, please explain.

3. In previous multi-swap local search approximation schemes, the approximation ratio is given as a function related to the swap size $t$ (i.e., in [1], the approximation is given as $(3+2/t)^2$). However, in this paper, the approximation ratio is determined by an inequality related to $\eta$ and $t$. Can the author provide the approximation ratio in the form similar to that of [1] such that one could figure out how large that swap size is enough for obtaining an approximation guarantee of 10.48.

4. Why are all the instances in the experimental setting small? In the recent results for clustering algorithms, instances of size over 1 million [2] or even 100 million [3] have been considered. How does the proposed multi-swap local search method scale on large datasets?

5. What are the performances of different local search algorithms within a certain time limitation (large enough for reaching the convergence)?

[1] Kanungo T, Mount D M, Netanyahu N S, et al. A local search approximation algorithm for k-means clustering[C]//Proceedings of the eighteenth annual symposium on Computational geometry. 2002: 10-18.

[2] Ren J, Hua K, Cao Y. Global Optimal K-Medoids Clustering of One Million Samples[J]. Advances in Neural Information Processing Systems, 2022, 35: 982-994.

[3] Matsui Y, Ogaki K, Yamasaki T, et al. Pqk-means: Billion-scale clustering for product-quantized codes[C]//Proceedings of the 25th ACM international conference on Multimedia. 2017: 1725-1733.



**Limitations:**

Since this is a theoretical paper, I don't think there is potential negative societal impact of this work.

---

> ### Author Rebuttal · Authors · 2023-08-09
>
> We thank the reviewer for their time and effort put into reviewing the paper. We start by addressing the weaknesses suggested by the reviewer and then we answer the specific questions.
>
> Answers to weaknesses:
>
> Weakness 1. Although the algorithm is an extension of the one in Lattanzi and Sohler, the analysis as carried out in [1] does not yield any low constant. We incorporated more elements into the analysis (some of which are borrowed from [1]) to obtain a better constant. We note that in the study of approximation algorithms, once the first milestone of a constant factor approximation is achieved, the goal becomes to minimize the gap between the lower and upper bound. As an example, a large body of work is devoted to improving the constant approximation factor for popular clustering objectives like $k$-median and $k$-means [3, 4, 5].
> Overall, we agree that the techniques used in the proofs in the main body of this paper are similar to those in [2]. However, we believe that (i) improving the constant for an interesting constant-factor approximation algorithm is per se interesting; (ii) the main technical novelty is in the 9+eps algorithm, which is presented extensively in the appendix.
>
> Weakness 2.
> We thank the reviewer for finding this minor issue in our proof. We submit in the rebuttal an updated version of the paper (we couldn't attach it to our response to the specific reviewer) where we implement a simple change. In particular, adding lines 192-194 and 218 to our manuscript fixes this issue, which are highlighted in blue to make it easily verifiable. The reviewer rightfully pointed out that not all optimal centers in $\tilde O$ were assigned to a swap. Indeed, this happens whenever $o \in \tilde O$ and $C[o] \not\in \tilde C$. We fixed this by creating 1-to-1 swaps between these “orphaned” centers and the leftover lonely centers. We have many enough of such lonely centers because by $|\tilde C| \geq |\tilde O|$.
>
> Weakness 3.
> (1) The reviewer raises a good point. We omitted this experiment given that the running times only differ by a small constant and the times reported suggest that our algorithms still outperform the 1-swap algorithm when run against a deadline. We will add an experiment where we compare the different algorithms against a set of deadlines that is set in the range [1x, …, 20x] the time for a single iteration of Lloyd's algorithm. In our revised version attached in the rebuttal we include this experiment for two of our datasets (KDD-BIO and KDD-PHY) in Figure 3. The experiment confirms our observation that our algorithm performing swaps of size greater than one still achieves better performance compared to the single-swap version of the algorithm despite performing fewer local search steps.
>
> (2) We note that this is a standard setting as in the case of the single-swap local search paper by Lattanzi and Sohler. Even for 50 iterations the algorithms appear to reach a point of diminishing returns in terms of quality improvements.
>
> (3) The main point of our work is to obtain a practical algorithm with good theoretical guarantees. We didn't try to optimize our implementation and we didn't employ any parallelization, which would lead to faster implementations. We believe the current setting of our experiment aligns well with the goal of our paper.
>
> We further note that the scalability comparison to the two papers mentioned by reviewer is not valid:
> The paper by Ren et al. only obtains reasonable times for datasets with millions of points for the case of $k=3$ and by applying parallelization using over a thousand cores, and still these times are in the order of hours.
> The paper by Matsui et al. considers only a heuristic approach, and shows that it is over an order of magnitude faster than plain $k$-means++. We iterate that our goal is to devise an efficient algorithm with provable approximation guarantees.
>
> Finally, we recall that even for the case of the 10-swap version of our algorithm, each iteration is faster than each iteration of Lloyd which is the standard post-processing algorithm applied in practice.
>
> Answers to questions:
>
> (1) See comment to Weakness 2 above.
>
> (2) Notice that the success probability indeed depends on $\delta$, as indicated by the notation $\Omega_{\delta}(1)$, which is introduced in Line 108. This notation is used to hide the depende on $\delta$ to hide unnecessary complexity in the claims, and is assumed to be a constant as stated in Theorem 3.
>
> (3) We agree that a simple formula to determine the approximation factor would be ideal. We note that solving equation (1) for $\eta$ would derive $\eta = \sqrt{5 p^2 + 4 p + 1}/p + 1/p + 1$, and hence the approximation factor in terms of $p$ and $\delta$ becomes $\left(  \sqrt{5 p^2 + 4 p + 1}/p + 1/p + 1 \right)^2 + \delta$, which is arguably not a very helpful formula to state in our main theorems. For this reason we preferred the simplicity of stating the theorems in terms of $\eta$, and just state that $p$ should be large enough. In the next version of our paper we can explicitly state the solution to equation (1) in a form of a footnote to facilitate the reader.
>
> (4) See comment on weakness 3 (2).
>
> (5) See comment on weakness 3 (1).
>
> [1] Kanungo T, Mount D M, Netanyahu N S, et al. A local search approximation algorithm for k-means clustering[C]//Proceedings of the eighteenth annual symposium on Computational geometry. 2002: 10-18.
>
> [2] Lattanzi, Silvio, and Christian Sohler. "A better k-means++ algorithm via local search." ICML 2019.
>
> [3] Jain, Kamal, and Vijay V. Vazirani. "Approximation algorithms for metric facility location and k-median problems using the primal-dual schema and Lagrangian relaxation." JACM 2001.
>
> [4] Ahmadian, Sara, et al. "Better guarantees for k-means and euclidean k-median by primal-dual algorithms." SICOMP 2019. FOCS17.
>
> [5] Cohen-Addad, Vincent, et al. "Improved approximations for Euclidean k-means and k-median, via nested quasi-independent sets". STOC 2022.

---

> > ### Comment · Reviewer_2sgw · 2023-08-21
> >
> > I would like to thank the reviewer for the clarification. However, the PDF provided by the authors used for rebuttal contains a revised version of the paper instead of just the figures and tables as required by the rebuttal instructions, which does not follow the rule for rebuttal. Thus, it is rather confused to me that whether the discussion should include the revised version of the PDF provided by the authors.

---

> > > ### Author Response · Authors · 2023-08-21
> > >
> > > We apologise for the confusion that was caused. Our intention was to facilitate the reviewer in validating our minor revision of the definition of ideal multi-swaps by marking the changes in blue in the context of the whole paper.
> > >
> > > Unfortunately, the system doesn't allow us to post a new PDF containing only the relevant changes. We note that the only relevant parts of the revised PDF (attached in our rebuttal comment to all reviewers) for this discussion are:
> > >
> > > * The changes for the minor issue with the definition of ideal multi-swaps: marked in blue in Lines 192-194, and 218, and
> > > * The requested experiment where we compare the algorithms against a fixed running time as suggested by the reviewer: in Figure 3, and the associated discussion in Lines 368-376

---

### Official Review · Reviewer_FVuS · 2023-07-02

**Soundness:** 4 excellent
**Presentation:** 3 good
**Contribution:** 4 excellent
**Rating:** 7
**Confidence:** 4

**Summary:**

This paper proposes a new k-means algorithm: multi-swap local search (MSLS) which combines local search and k-means++, to achieve a constant approximation guarantee with efficient time complexity. Specifically, the local search framework includes a step that selects alternative centers for optimizing the cost function, and this paper selects those centers using the $ D^2 $-sample from k-means++  (without updating costs).

The authors prove that the random candidate centers obtained in this way lead to an improvement over the cost. By iterating on this step, MSLS will return a constant approximation result.

The author also leverage ideas from coreset and  dimensionality reducion to propose a more efficent $ 9+\varepsilon $-approximation algorithm within the local search framework.

**Strengths:**

- The paper is well written. All sources of motivations and techniques involved in this work are clearly presented, making it easy to follow the key ideas of the algoirthm and the related works. It does make a novel contribution to the classic problem.
- The idea behind the MSLS algorithm is natural and the proofs are very clear. Although the author refer techniques from the previous work, the approximation analysis is non-trivial and results in very tight ratios.
- The MSLS algorithm is simple and easy to implement. The experiments are comprehensive. This work is valuable in both theory and practice.

**Weaknesses:**

- the time complexity is $ \tilde{O} (nd k^p) $, which grows exponentially w.r.t. $p$.

**Questions:**

- Both k-means++ and local search are not limited to the Euclidean space. Can thm3 be generalized to the metric space? I notice that the analysis are influenced by [1] and there exists mean computing ($ \mu(\cdot) $) that prevents such generalization. However, [2] gives a constant approximation for k-means in the metric space. Therefore, I am interested in what the current analysis method lacks for the generalization and whether such limitation can be overcomed?



[1] A Local Search Approximation Algorithm for k-Means Clustering


[2] Simpler Analyses of Local Search Algorithms for Facility Location

**Limitations:**

-

---

> ### Author Rebuttal · Authors · 2023-08-09
>
> We thank the reviewer for their time and effort put into reviewing the paper.
>
> We would like to clarify that some exponential dependence on p is likely needed for any local-search based algorithm. In particular, if one could devise an exhaustive local search algorithm with polynomial dependence on $p$, then setting $p=k$ one would get a polynomial-time algorithm to solve this NP-hard problem that is known to be hard to approximate within a factor of 1.06 [1]. Notice that our algorithm is a $(1+\epsilon)$ factor away from the approximation achieved by the exhaustive local search by Kanungo et al. which is analyzed for a fixed number of swaps. This means that if we were able to devise an algorithm with sub-exponential dependence on $p$, for $p=k$, we would hope to achieve a $(1+\epsilon)$ factor approximation to the k-means problem, which is NP-hard. However, it is a very interesting and open question to achieve a dependence on $c^{f(p)}$, for a constant $c$.
>
> It is indeed an interesting open question on whether one can achieve a similar result in general metric spaces. We tried to extend the analysis from [2] to $k$-means but we find one main obstacle: their projection lemma (Lemma 2.4 in their arxiv version) does not hold for $k$-means because distances squared do not satisfy triangle inequality. The corresponding lemma in our work is Lemma 8 whose proof crucially relies on Lemma 1, which in turn only works for euclidean spaces.
>
> [1] Vincent Cohen-Addad and Karthik C. S. Inapproximability of clustering in lp metrics. In FOCS'19.
> [2] Gupta, Anupam, and Kanat Tangwongsan. "Simpler analyses of local search algorithms for facility location." arXiv preprint arXiv:0809.2554 (2008).

---

> > ### Comment · Reviewer_FVuS · 2023-08-17
> >
> > Thanks for your response. I would like keep my assessment unchanged.

---

### Official Review · Reviewer_swvB · 2023-07-06

**Soundness:** 3 good
**Presentation:** 3 good
**Contribution:** 4 excellent
**Rating:** 7
**Confidence:** 4

**Summary:**

The following results are given in the paper:
1. A tighter analysis of the local search algorithm of Lattanzi and Sohler. The paper shows a constant approximation guarantee for their algorithm.
2. The paper extends the approach of Lattanzi and Sohler to multi-swap local search (where more than one point is swapped in every iteration). Using this approach, an approximation guarantee of 10.48 is obtained.
3. The paper also gives a (9+\eps) approximation. Such a 9-approximation algorithm using local search is already known (https://www.sciencedirect.com/science/article/pii/S0925772104000215). The algorithm given in this paper has a slight running time advantage.

The algorithm is a local search algorithm. The justification of the title is that the swap-in points in the local search strategy are obtained by D^2 sampling (i.e., the sampling strategy used in k-means++).

**Strengths:**

1. Knowing a better analysis for an already published algorithm is good.
2. Extension to multi-swap and its analysis giving better approximation guarantee is interesting.
3. Matching the best possible approximation (possible using local search) using a slightly better running time is also interesting.
4. The experimental section shows that the local search iterations do better than Lloyd's. This is interesting and can be of practical use. It would have been better if the running time comparison was also given to make the tradeoffs more visible.

**Weaknesses:**

1. The paper looks good from the point of view of extensions to the Lattanzi and Sohler work. However, from the point of view of developments in local search algorithms (which is essentially what the paper is), the improvement over (https://www.sciencedirect.com/science/article/pii/S0925772104000215) seems incremental. Even the proof techniques are similar.
2. There are some discrepancies in the experimental section. For example, Figure 2 says "first row...", but there is none. I did not find the running time for the local search iterations (even MSLS-G).

**Questions:**

- Some of the other queries are mentioned above within the other fields.

**Limitations:**

This work is mostly theoretical. There are no negative societal impacts.

---

> ### Author Rebuttal · Authors · 2023-08-09
>
> We thank the reviewer for their time and effort put into reviewing the paper.
>
> We acknowledge that the proof technique is similar to the one by Kanungo et al. on a high level. However, the exhaustive local search technique of their paper leads to their algorithm being prohibitive for practical applications. The inefficiency of the Kanungo et al. paper serves as the main motivation of our work that sharply improves the running time to $O(n\cdot d \cdot poly(k))$ for an algorithm with approximation ~10.48 for any constant number of swaps, which is further implementable and gives rise to a very efficient heuristic. Our more elaborate $9+\epsilon$ approximation algorithm essentially shows that one can match the approximation factor of Kanungo et al. by leveraging coresets and obtaining an algorithm that is faster for a wide range of values of k.
>
> By "first row", we refer to the first row of figures, as the two top figures. We kindly note that the running times of the algorithms (per iteration) are reported in the legends of Figure 2 following the algorithm names. However, we will clarify both these points in the final version of the paper.

---

> > ### Comment · Reviewer_swvB · 2023-08-11
> >
> > Thanks for your response. I do not have any more questions.

---

### Author Rebuttal · Authors · 2023-08-09

While we respond to each reviewer individually, we attach here a revised version of our paper where we incorporate things discussed with reviewer 2sgw.

We note that while this attached version is slightly longer than 9 pages, we only use it to refer to it in this rebuttal phase, and will be made to abide to all formatting guidelines in the final version.

---

### Decision · Program_Chairs · 2023-09-21

**Decision:**

Accept (poster)

**Comment:**

This paper is a difficult case as one very thoughtful reviewer has raised valid concerns that seem to be largely addressed, though the authors did not follow the correct format in uploading a rebuttal version of the changes. These concerns are in a sense outweighed by the positive consensus among other reviewers, who relatively provide very little information behind their assessments.